# Depolymerization mechanisms and closed-loop assessment in polyester waste recycling

Jingjing Cao [1,5], Huaxing Liang[1,5], Jie Yang[2], Zhiyang Zhu[1], Jin Deng [2] ✉, Xiaodong Li [3] ✉, Menachem Elimelech [4] ✉ & Xinglin Lu [1] ✉

Alcoholysis of poly(ethylene terephthalate) (PET) waste to produce monomers, including methanolysis to yield dimethyl terephthalate (DMT) and glycolysis to generate bis-2-hydroxyethyl terephthalate (BHET), is a promising strategy in PET waste management. Here, we introduce an efficient PET-alcoholysis approach utilizing an oxygen-vacancy ($V_O$)-rich catalyst under air, achieving space time yield (STY) of 505.2 $g_{DMT}\cdot g_{cat}^{-1}\cdot h^{-1}$ and 957.1 $g_{BHET}\cdot g_{cat}^{-1}\cdot h^{-1}$, these results represent 51-fold and 28-fold performance enhancements compared to reactions conducted under $N_2$. In situ spectroscopy, in combination with density functional theory calculations, elucidates the reaction pathways of PET depolymerization. The process involves $O_2$-assisted activation of $CH_3OH$ to form $CH_3OH^*$ and $OOH^*$ species at $V_O$-$Zn^{2+}$-$O$-$Fe^{3+}$ sites, highlighting the critical role of $V_O$-$Zn^{2+}$-$O$-$Fe^{3+}$ sites in ester bond activation and C–O bond cleavage. Moreover, a life cycle assessment demonstrates the viability of our approach in closed-loop recycling, achieving 56.0% energy savings and 44.5% reduction in greenhouse-gas emissions. Notably, utilizing PET textile scrap further leads to 58.4% reduction in initial total operating costs. This research offers a sustainable solution to the challenge of PET waste accumulation.

Poly(ethylene terephthalate) (PET), a prominent type of petroleum-based polyester plastic, is widely used in producing bottles, packaging materials, and textiles[1,2]. Such wide applications result in PET waste (-70 million tons per year), accounting for -12% of global plastic waste[3,4]. Owing to PET material's inert nature, PET waste natural degradation in the environment is extremely slow[5], leading to potentially adverse environmental impacts and risks for the direct disposal of PET waste into landfills or water environments[6–8]. Consequently, the development of sustainable strategies for PET waste management is of paramount importance[9].

Chemical recycling is a promising approach that depolymerizes PET into monomer units for subsequent polymerization or selective transformation into high-value chemicals. To date, various chemical recycling methods have been demonstrated, including upcycling (pyrolysis[10], hydrosilylation[11], enzymatic[12,13], and hydrogenolysis[14–16]) and cycling (glycolysis[17–20], hydrolysis[21–23], methanolysis[24–26], and ammonolysis[27]). Alcoholysis is a process that utilizes alcohol as a solvent to depolymerize PET into monomers. The efficacy of this process largely relies on the design of the catalyst. Previous studies on this process typically involved the use of catalysts like homogeneous metal salts, ionic liquids, enzymes, and other catalysts[28–30]. While these catalysts achieve adequate efficiency, challenges such as difficulty in separating the product from the catalyst, leaching of metal ions, or catalyst instability remain prevalent. Consequently, developing a green, efficient, and sustainable catalyst for alcoholysis to convert PET into monomers is imperative.

[1]CAS Key Laboratory of Urban Pollutant Conversion, Department of Environmental Science and Engineering, National Synchrotron Radiation Laboratory, University of Science and Technology of China, Hefei, China. [2]CAS Key Laboratory of Urban Pollutant Conversion, Anhui Province Key Laboratory of Biomass Clean Energy, Department of Applied Chemistry, University of Science and Technology of China, Hefei, China. [3]Max Planck Institute of Microstructure Physics, Weinberg 2, Halle, Germany. [4]Department of Chemical and Environmental Engineering, Yale University, New Haven, CT, USA. [5]These authors contributed equally: Jingjing Cao, Huaxing Liang. ✉e-mail: dengjin@ustc.edu.cn; xiaodong.li@tu-dresden.de; menachem.elimelech@yale.edu; xinglinlu@ustc.edu.cn

A recent study introduced the concept of creating a solid-solid interface between plastic and catalyst, allowing methanolysis of the polymer through methanol vapor[26]. This innovative approach to interfacial catalysis opens up new avenues for the depolymerization of plastic waste. Additionally, other studies also demonstrate the promise of using oxygen vacancy-metal oxide in catalyzing the depolymerization of PET to produce BHET monomers[31–33]. For instance, defect-rich CeO$_2$ nanoparticles modified with KH550 have been used for the glycolysis of PET, at the critical boiling point (197 °C) of ethylene glycol, a PET conversion rate of 98.6% with a BHET yield of 90.3% was achieved using this approach[34].

Despite these encouraging findings, there are limited efforts on validating catalysts for real polyester waste, such as cotton blends, textiles, and undegraded carpets, leaving several challenges unaddressed[35–37]. First, most articles focus solely on catalyst design and PET depolymerization efficiency, with few investigating the pathways and mechanisms of PET depolymerization. This knowledge gap exists because the alcoholysis process involves high-temperature, solid-liquid-solid complex interface reactions among catalysts, polymers, and solvents. Moreover, there is a lack of research on the interactions between catalysts and solvents, solvents and polymers, and catalysts and polymers during the reaction process. Second, few studies examine the recycling of actual PET plastics or textiles, which contain additives like pigments and plasticizers that can affect the catalytic efficiency. Third, while PET recycling aims to recover waste, comprehensive life cycle evaluations, as well as economic and technical analysis of the entire recycling process, are scarce. Therefore, innovative catalyst design and deeper insights into reaction mechanisms, are critically needed for closed-loop recycling of real PET waste.

Here, we synthesize an oxygen vacancy ($V_o$)-rich Fe/ZnO nanosheets (NSs) catalyst for polyester plastic depolymerization. The average mass activity (space time yield, STY) of $V_o$-rich Fe/ZnO NSs reached 957.1 g$_{BHET}$ g$_{cat}^{-1}$ h$^{-1}$ (glycolysis) at 180 °C for 1 h, and 505.2 g$_{DMT}$ g$_{cat}^{-1}$ h$^{-1}$ (methanolysis) at 160 °C for 1 h, surpassing performance reported in the literature (Fig. 1b). In situ FTIR and isotope-labeling results indicate that the efficient PET depolymerization over $V_o$-Zn$^{2+}$–O–Fe$^{3+}$ sites originate from a series of processes compromising multiple O–O bond activation, CH$_3$OH dehydrogenation, nucleophilic attack, C=O activation, and cleavage of the C–O bond reaction pathway. Furthermore, we demonstrate a closed-loop production of commercial textiles using recycled DMT monomer. This catalyst exhibits superior tolerance and high catalytic activity in depolymerization of various real polyester wastes. A life cycle assessment (LCA) of this method reveals 56.0% energy savings and a 44.5% reduction in greenhouse gas emissions compared to conventional methods for plastic production, highlighting the promise of the developed strategy for the sustainable management of PET wastes.

## Results

### Catalytic depolymerization of polyester

Our experiments conducted the glycolysis process at 180 °C for 1 h under air. PET was completely converted to BHET (approaching 95.5% yield), which was detected by high-performance liquid chromatography (HPLC). Our $V_o$-rich Fe/ZnO NSs catalyst could convert PET into BHET with much higher activity (957.1 g$_{BHET}$·g$_{cat}^{-1}$·h$^{-1}$) than the catalysts in the literature (Fig. 1a left panel and Supplementary Table 1). Considering different reaction conditions used in previous studies, we divided the performance data of glycolysis into three categories (Fig. 1a): (1) glycolysis at the same temperature; (2) glycolysis at the critical boiling point of ethylene glycol; and (3) glycolysis at the melting point temperature of PET. Similarly, PET flake wastes could be completely converted through methanolysis in 1 h at 160 °C to obtain DMT with 99% yield (>99.5% purity) detected by gas chromatography. In particular, the presence of $V_o$-rich Fe/ZnO NSs catalyst results in a DMT formation rate of 505.2 g g$^{-1}$ h$^{-1}$, which is an order of magnitude

higher than that of the reported catalysts for methanolysis (Fig. 1b right panel and Supplementary Table 2). Considering different reaction conditions used in previous studies, the performance of methanolysis has been divided into four categories (Fig. 1b): (1) mixed solvent (tetrahydrofuran/chloroform)-assisted; (2) homogeneous catalysis; (3) heterogeneous catalysis; and (4) heterogeneous layer-assisted. The heterogeneous layer-assisted process illustrates the space time yield (g$_{PET}$ g$_{cat}^{-1}$ h$^{-1}$). Further optimization of reaction conditions, including temperature, time, and catalyst dosage (Supplementary Table 3), results in a yield of over 99% DMT with 99% purity (Fig. 1c$_1$). Notably, under air conditions, methanolysis of other wastes made of pure PET, such as PET slices (Fig. 1c$_2$) and bottle tablets (Fig. 1c$_3$) also led to >99% yield of DMT using $V_o$-rich Fe/ZnO NSs catalyst. In contrast, under N$_2$ atmosphere, the PET conversion rate is less than 10%, with the resulting products primarily consisting of oligomers (Supplementary Fig. 5).

Besides the pure polyester wastes, we further conducted methanolysis on mixed polyester wastes (PET/PC particles, Fig. 1c$_4$ and Supplementary Fig. 6). For PET/PC particles that both compositions are modes of polyesters, we proposed a selective chemical depolymerization strategy for recycling both compositions. First, when the reaction temperature is 120 °C, polycarbonate (PC) was completely converted into bisphenol A (98% yield, >99.5% purity within 1 h). While PET particles hardly underwent any reaction under such temperature even extending the reaction time to 24 h (Supplementary Table 3). The comparison of the results suggests a significant thermodynamic difference in enabling selective chemical depolymerization. Further increasing the temperature to 160 °C enabled efficient depolymerization of PET to generate DMT. Such a result demonstrates the utilization of temperature differences for selective and sequential depolymerization of polyester mixture wastes (like PC/PET) using $V_o$-rich Fe/ZnO NSs catalyst[18,38,39]. This approach enables the selective chemical depolymerization of PET/PC mixed plastics to yield monomers, i.e., DMT (from PET) and BPA (from PC), effectively overcoming the challenges associated with mechanical sorting and separating mixed depolymerization products.

We assessed the tolerance of $V_o$-rich Fe/ZnO NSs in catalytic methanolysis of polyester in composite plastic wastes, including PET/PP packing boxes (~50 wt% PET, Fig. 1c$_5$), PET/PE thin films (~30 wt% PET, Fig. 1c$_6$), and PET color bottles (mixture of PET, PP, PE, and pigment, ~95 wt% PET, Fig. 1c$_{7-8}$). A small amount (1 g) of sliced pieces (~2.0 cm in size) of these wastes were added into methanol with 1 wt% catalyst and stirred at 160 °C. The catalytic reactions lead to 99%, 99%, and 98% yields of DMT for PET/PP packing boxes, PET/PE thin films, and PET bottles, respectively, demonstrating the remarkable performance of the synthesized catalyst. Notably, this catalyst also exhibits good activity in depolymerization of a large amount (40 g) of plastic pieces (~2.0 cm in size), leading to the isolation of 37.0 g DMT with high purity (Supplementary Table 3). Moreover, the $V_o$-rich Fe/ZnO NSs catalyst in polyester depolymerization possesses structural stability, and it maintained high activity and selectivity after 5 cycles. After the regeneration of the catalyst (catalyst regeneration conditions provided in Supplementary methods), the catalytic activity can still return to its initial value (Supplementary Table 3). In the XRD patterns and TEM images (Supplementary Fig. 7), the regenerated $V_o$-rich Fe/ZnO NSs catalyst showed similar nanosheet structures and the same phase patterns as the original sample.

### Characteristics of $V_o$-rich Fe/ZnO NSs catalyst

$V_o$-rich Fe/ZnO NSs were synthesized by adopting an organic base-assisted thermal decomposition strategy. Assembly of zinc chloride, ferric chloride, and L-alanine in an ethanolamine solution driven by oriented attachment interactions results in the formation of Fe-Zn precursors. Pyrolysis of the Fe-Zn precursor compounds led to the formation of ultrathin $V_o$-rich Fe/ZnO NSs catalyst (Fig. 2a). To shed light on the actual contributions of vacancies in PET depolymerization,

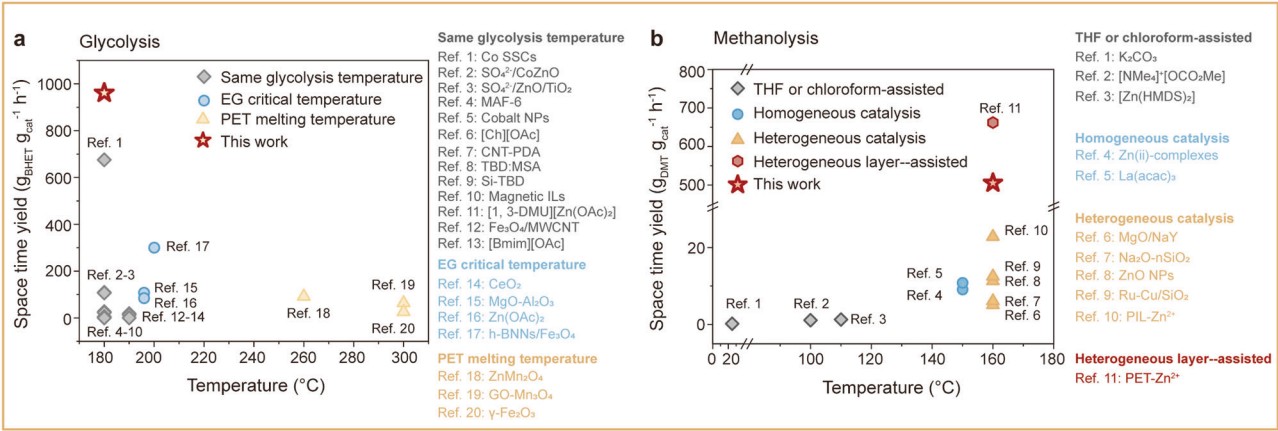

**c** Catalytic depolymerization of polyester wastes

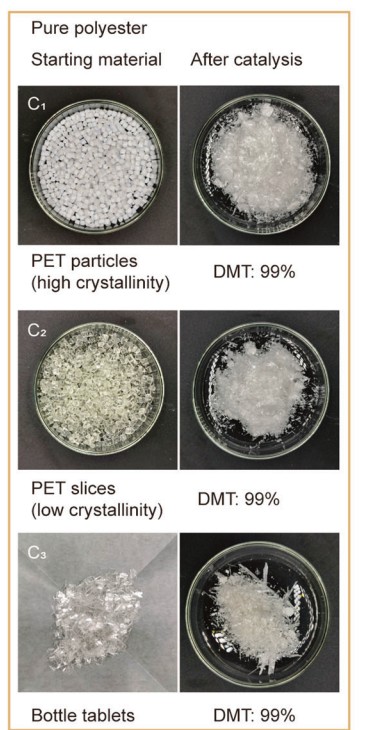
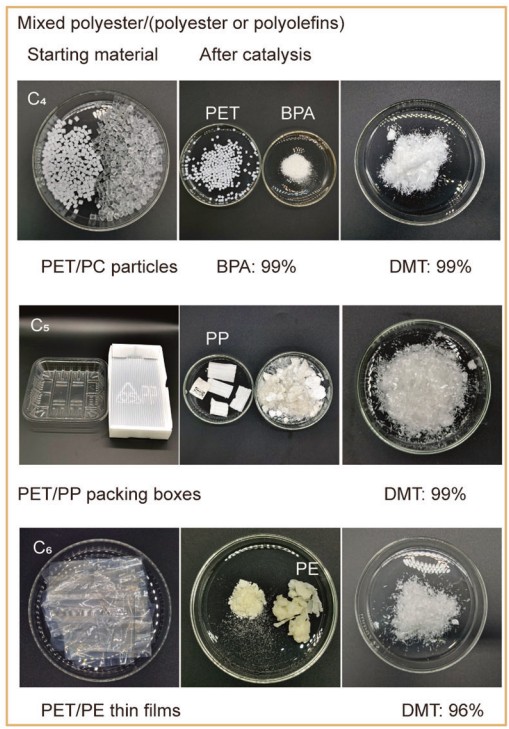
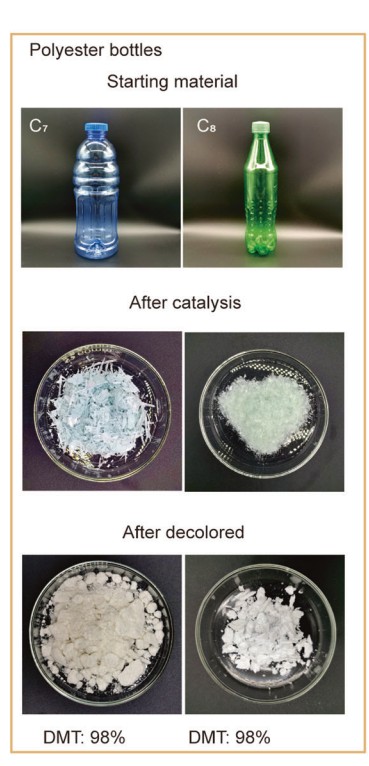

**Fig. 1 | Catalytic depolymerization of polyester.** Diagrams of **a** PET glycolysis and **b** methanolysis over $V_o$-rich Fe/ZnO NSs and their catalytic performance in comparison to catalysts in the literature. Reaction rate is estimated using the following equation: STY= $\frac{m_{monomers}}{m_{cat} \times t}$. **c** Catalytic depolymerization of mixed polyester wastes. Experiments were carried out under an air atmosphere. DMT yields were detected by gas chromatography.

we also synthesized oxygen vacancy-poor ($V_o$-poor) Fe/ZnO NSs and bulk ZnO materials for comparison (details provided in the Method section).

The resulting $V_o$-rich Fe/ZnO NSs exhibit a 2D nanosheet structure with an average thickness of 2.0 nm using atomic force micrography (AFM) (Fig. 2b). This thickness corresponds to the total thickness of a five-unit cell ZnO slab. We also observed 2D nanosheet morphology and uniform distribution of Fe element in $V_o$-rich Fe/ZnO NSs (Fig. 2c). In X-ray diffraction (XRD), both $V_o$-rich Fe/ZnO NSs and $V_o$-poor Fe/ZnO NSs display the characteristics pattern attributable to hexagonal ZnO (JCPDS no.79-0208), indicating unchanged phase structure of ZnO with the low contents of Fe (Supplementary Fig. 8)[40]. In contrast, the bulk ZnO has a wurtzite structure (JCPDS no. 36-1451) that is distinctly different from the structure of the synthesized $V_o$-rich Fe/ZnO NSs.

Aberration-corrected high-angle annular dark-field scanning transmission electron microscopy (HAADF-STEM) was employed to reveal the fine structures of $V_o$-rich Fe/ZnO nanosheets (NSs). As depicted in Fig. 2d, slight lattice disorders have been locally observed in the nanosheets, likely stemming from vacancies induced by the unsaturated coordination of metal atoms. The $V_o$-rich Fe/ZnO NSs exhibit interplanar spacings of 0.283 nm, corresponding to the distances of the (100) planes of ZnO (Supplementary Fig. 9). Energy-dispersive X-ray spectroscopy (EDS) and inductively coupled plasma (ICP) analysis indicate that the iron doping content of the nanosheets is 0.7% and 0.8%, respectively (Supplementary Fig. 10). This iron concentration is close to the theoretical value (~1%) determined from the weight ratio used during the catalyst synthesis. Unpaired electrons on these defects could be further assessed using electron spin resonance (ESR) (Fig. 2e). Bulk ZnO without vacancies showed an intensity at a $g$ value of 1.950, attributed to electron trapping at the lattice of Zn sites ($V_{Zn}$)[41]. In comparison, $V_o$-rich Fe/ZnO NSs have abundant oxygen vacancies, as evidenced by their strong ESR intensity at a $g$ value of 2.002[42]. We used X-ray photoelectron spectroscopy (XPS) to

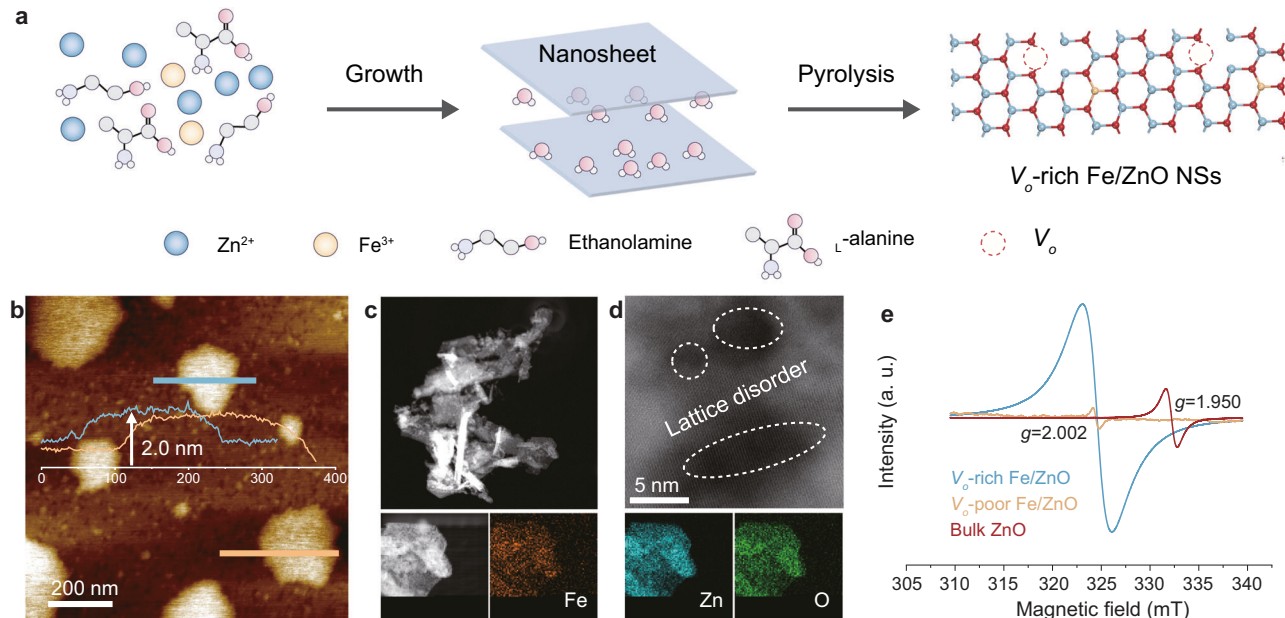

**Fig. 2 | Synthesis and structural characteristics of $V_o$-rich Fe/ZnO NSs.**
**a** Schematic illustration of the synthesis of $V_o$-rich Fe/ZnO NSs. **b** AFM images and
height distributions of $V_o$-rich Fe/ZnO NSs. **c** HAADF-STEM image and
corresponding EDS mapping of $V_o$-rich Fe/ZnO NSs. **d** Aberration-corrected
HAADF-STEM image of $V_o$-rich Fe/ZnO NSs. **e** ESR profiles of bulk ZnO, $V_o$-poor Fe/
ZnO NSs, and $V_o$-rich Fe/ZnO NSs.

determine the valence state of $V_o$-$Zn^{2+}$–O–$Fe^{\delta+}$. In the Zn 2p region
(Supplementary Fig. 11a), Zn $2p_{1/2}$ and Zn $2p_{3/2}$ peaks appear at 1044.7
and 1021.6 eV, respectively, indicative of the +2 oxidation state of Zn. In
the Fe 2p region (Supplementary Fig. 11b), Fe $2p_{1/2}$ and Fe $2p_{3/2}$ peaks
appear at 724.9 and 711.3 eV, indicative of the +3 oxidation state of Fe.
As such, $V_o$-$Zn^{2+}$–O–$Fe^{\delta+}$ was determined to be $V_o$-$Zn^{2+}$–O–$Fe^{3+}$. Addi-
tionally, in the O 1 s region (Supplementary Fig. 11c), the peaks at
531.4 eV and 529.7 eV correspond to the O atoms in the vicinity of
oxygen vacancies and the lattice oxygen of Zn–O–Fe, respectively[43].
These findings indicate that the addition of iron plays an important
role in the effect of increasing oxygen vacancy density. This unique
structure ensures that the active sites and abundant vacancies are
highly desirable for catalyzing the methanolysis of PET.

**Evolution of $V_o$-rich Fe/ZnO NSs catalyst structure**
The catalytic performance and mechanisms of $V_o$-rich Fe/ZnO NSs can
be better evaluated by examining their defect structures. We con-
ducted theoretical investigations on $V_o$-rich Fe/ZnO surfaces for PET
depolymerization. The mode started with a ZnO (100) surface (Sup-
plementary Fig. 12), constructed using a (4 × 4 × 4) supercell consisting
of four atom layers. Then, we introduced a Fe dopant by replacing one
Zn atom with one Fe atom (schematically illustrated in Supplementary
Fig. 13). The creation of oxygen vacancies involved the direct removal
of surface oxygen atoms (detailed mode provided in the Supplemen-
tary Information). Notably, the oxygen vacancy sites on the top slab of
ZnO (100) surfaces were more readily generated (ΔE = 3.55 eV) com-
pared to the second layer slab (Fig. 3a and Supplementary Table 4). To
explore the most stable structure of Fe atom-doped ZnO (100) with
oxygen vacancies, we calculated the formation energy of oxygen
vacancies at various positions (D1 to D11, shown in Supplementary
Fig. 13). The results demonstrate that the fifth configuration (D5) has
the lowest defect formation of 3.535 eV, which is even lower than that
in the pure ZnO slab.

In situ electron spin resonance (ESR) spectra were conducted to
monitor the evolution of $V_o$-rich Fe/ZnO NSs during different reac-
tion temperature stages under different atmospheres (air or nitro-
gen, Fig. 3b). The ESR signal at g = 2.002, corresponding to oxygen
vacancy, could be attributed to the $V_o$-$Zn^{2+}$–O–$Fe^{3+}$ structure, which

was from the rich-$V_o$ induced by the low coordination structure[43].
Notably, the intensity of the peak shows a progressive increase with
rising reaction temperature under air, indicating an increase in oxy-
gen vacancy density. In contrast, the intensity of the peak
progressively decreases with rising reaction temperature under $N_2$,
implying a decreased density of oxygen vacancy on $V_o$-rich Fe/ZnO
NSs. Taken together, these results demonstrate the activation of $O_2$
(in air) is an important factor in increasing the density of oxygen
vacancies.

Besides the role of the oxygen vacancy sites, we conducted in situ
experiments using attenuated total reflectance (ATR) Fourier trans-
form infrared spectroscopy (FTIR) to elucidate reaction mechanisms
of the solvent (i.e., methanol, $CH_3OH$) in activation of $V_o$-rich Fe/ZnO
NSs surface (Fig. 3c). The in situ measurements were operated by
gradually increasing the temperature from 25 to 160 °C and collecting
spectra at -10 °C intervals. During the reaction (Fig. 3c), we observed
the $V_{as}$(OH) signal at 3600–3080 $cm^{-1}$, corresponding to the hydroxyl
group, gradually decreased, suggesting the activation of a primary
hydroxyl group in methanol[43]. Moreover, with increasing temperature
(Fig. 3c), we observed the transformation of the hydroxyl group (at
1030 $cm^{-1}$) into triply bridged hydroxyl groups (at 1004, 1030, and
1056 $cm^{-1}$). Such results indicate that the $\beta$ interaction (1056 $cm^{-1}$) and
$\gamma$ interaction (1004 $cm^{-1}$), attributed to the alkoxy bond between pri-
mary hydroxyl and metal oxides, gradually increase[44,45]. Compared
with the $N_2$ atmosphere (Supplementary Fig. 14), the intensity of the
hydroxyl group (1030 $cm^{-1}$) adsorbed on the catalyst surface remained
unchanged, and the characteristic peaks of the $\beta$ interaction
(1056 $cm^{-1}$) and $\gamma$ interaction (1004 $cm^{-1}$) did not appear, indicating
that methanol was not activated by catalyst under nitrogen
atmosphere.

We propose a $V_o$-$Zn^{2+}$–O–$Fe^{3+}$ site to analyze its electronic inter-
play of Zn, Fe ions, and oxygen vacancy. In $V_o$-$Zn^{2+}$–O–$Fe^{3+}$sites, the
three unpaired electrons in the $\pi$-symmetry ($t_{2g}$) $d$–orbitals of $Fe^{3+}$
interact with the bridging $O^{2-}$ via $\pi$-donation. In contrast, the domi-
nant interaction between the fully occupied $\pi$-symmetry ($t_{2g}$)
$d$–orbitals of $Zn^{2+}$ and the bridging $O^{2-}$ is electron-electron repulsion,
leading to partial electron transfer from $Zn^{2+}$ to $Fe^{3+}$ (Supplementary
Fig. 15)[46,47]. The basic oxygen vacancies on the surface of $V_o$-rich Fe/

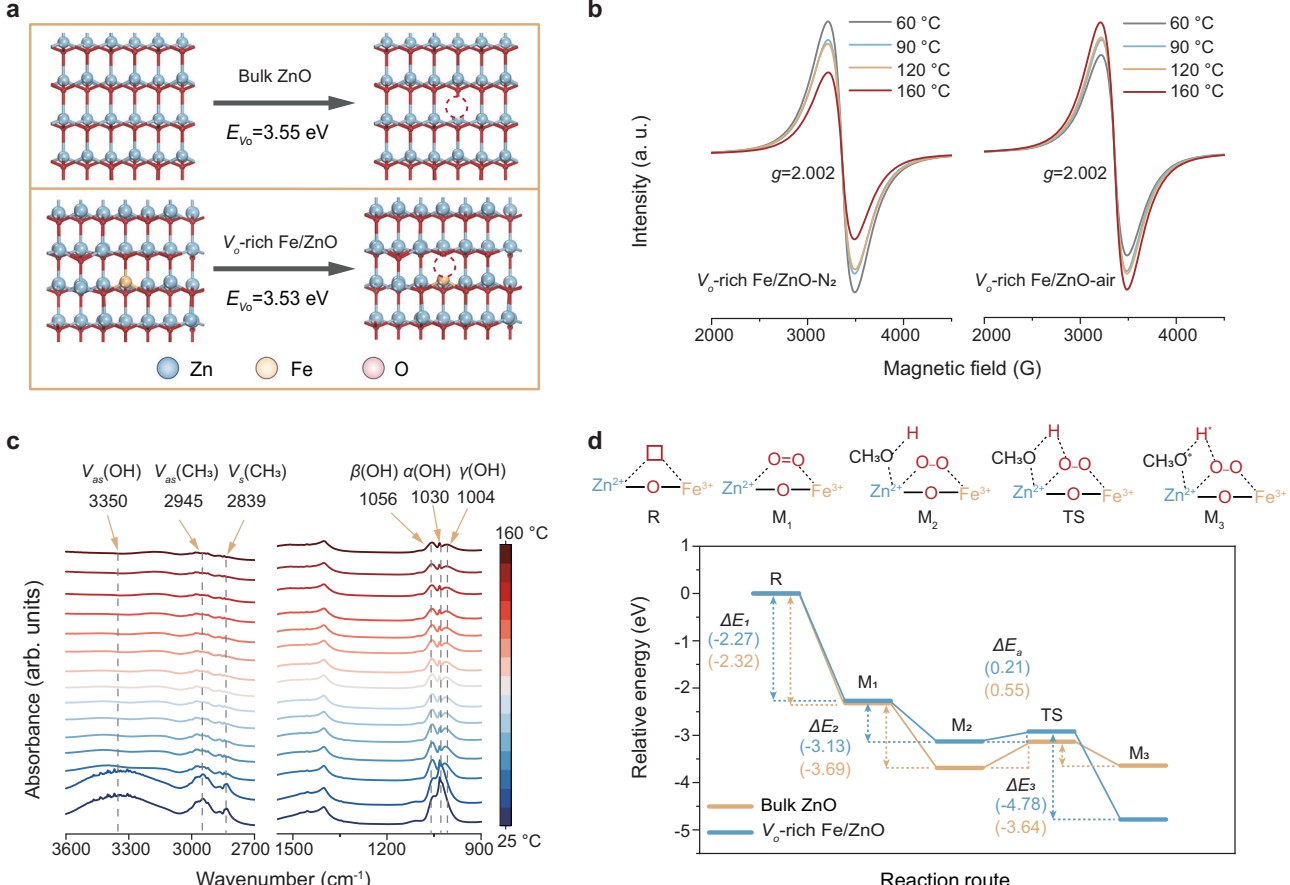

**Fig. 3 | Structural Evolution of $V_o$-rich Fe/ZnO NSs Catalyst. a** Oxygen vacancy formation energy of bulk ZnO and $V_o$-rich Fe/ZnO NSs. **b** ESR profiles of $V_o$-rich Fe/ZnO NSs under $N_2$ and air atmospheres. **c** In situ attenuated total reflectance (ATR) infrared spectra of the $CH_3OH$ to $CH_3OH^*$ process over $V_o$-rich Fe/ZnO NSs under air atmosphere. **d** Free energy profiles for the conversion of $CH_3OH$ to $CH_3OH^*$ on bulk ZnO and $V_o$-rich Fe/ZnO NSs.

ZnO NSs can serve as adsorption sites for oxygen molecules and methanol.

In addition, to assess the change of methanol activation energy barrier adsorbed on the $V_o$-rich Fe/ZnO NSs surface, we constructed an energy diagram of the reaction pathways (Fig. 3d), including the formation of $O_2 \rightarrow O_2^*$, as well as the determination of the activation energies of $CH_3OH + O_2^* \rightarrow OOH^* + CH_3OH^*$. $V_o$-$Zn^{2+}$–O–$Fe^{3+}$ localized oxygen vacancy structure (**R**) anchors an oxygen molecule to form **$M_1$** species. The hydroxyl hydrogen of the methanol ($CH_3OH$) molecule adsorbed on **$M_1$** is activated by oxygen molecules to form **$M_2$** species. Then $CH_3OH$ of **$M_2$** and $V_o$-$Zn^{2+}$–O–$Fe^{3+}$ form a transition state (**TS**) with a metal alkoxy bond, finally leading to the formation of the **$M_3$** structure of the nucleophilic species. Notably, the activation energies of the transition state (**TS**) activation revealed that bulk ZnO (0.55 eV) (Supplementary Fig. 16 and Supplementary Table 6). In contrast, $V_o$-rich Fe/ZnO NSs exhibit low activation energies (0.21 eV) (Fig. 3d, Supplementary Fig. 17 and Supplementary Table 7). This result indicates that the $V_o$-rich Fe/ZnO NSs possess a higher activity for $O_2 \rightarrow O_2^*$ and $CH_3OH + O_2^* \rightarrow OOH^* + CH_3OH^*$ species, essential for the subsequent C–O disconnection of PET depolymerization. These findings align with the observations using in situ FTIR, underscoring that $V_o$-rich Fe/ZnO NSs featuring $V_o$-$Zn^{2+}$–O–$Fe^{3+}$ are highly effective in catalyzing the activation of $CH_3OH$.

### Reaction pathways and catalytic mechanisms

Crystalline domains are less susceptible to PET depolymerization than amorphous domains. During the depolymerization process (160 °C), the melting temperature ($T_m$) of PET decreased from 247 °C

(for pristine PET) to 224 °C after 20 min (Fig. 4a). This reduction in $T_m$ indicates an increased portion of the amorphous domain (small molecules or chain ends) in the PET matrix. Such changes in the thermal properties are likely to have a profound impact on catalyst activity and performance[48]. The weight-average molecular weight ($M_w$) of PET, as determined by gel permeation chromatography (GPC), dropped from 59.1 $kDa$ (for pristine PET) to 12.9 $kDa$ after 5 min at 160 °C (Fig. 4b), which was associated with the presence of oligomers ($M_w$ ranges highlighted in the yellow shaded box in Fig. 4b). Upon extending the reaction time to 20 min, $M_w$ further decreased to 3.4 $kDa$, indicating efficient scission of C–O bonds in polymeric chains of PET. Previous studies suggest that during PET depolymerization, random scission of C–O bonds in the amorphous domain, together with insufficient cracking of the highly crystalline domains, results in a large dispersity of $M_w$[49]. In contrast, in our system, the presence of $V_o$-rich Fe/ZnO NSs in PET depolymerization led to PET with narrowed and progressively reduced dispersity, suggesting efficient cracking of large polymer molecules. Notably, as the PET depolymerization proceeded, the polymeric residues exhibited wider PDI and smaller $M_w$ values (PDI = 4.02 and $M_w$ = 13.8 $kDa$) compared to pristine PET (PDI = 1.47 and $M_w$ = 64.9 $kDa$) ($M_w$ range highlighted in the blue shaded box in Fig. 4b). The results suggested that PET interfacial catalysis reaction can be divided into two stages (Supplementary Fig. 18). In the initial stage, depolymerization proceeds slowly, with the fracture of the amorphous part of the polymer chain playing the dominant role. In the latter stage, the depolymerization rate of PET accelerates, with an increase in active species and the appearance of monomers playing a

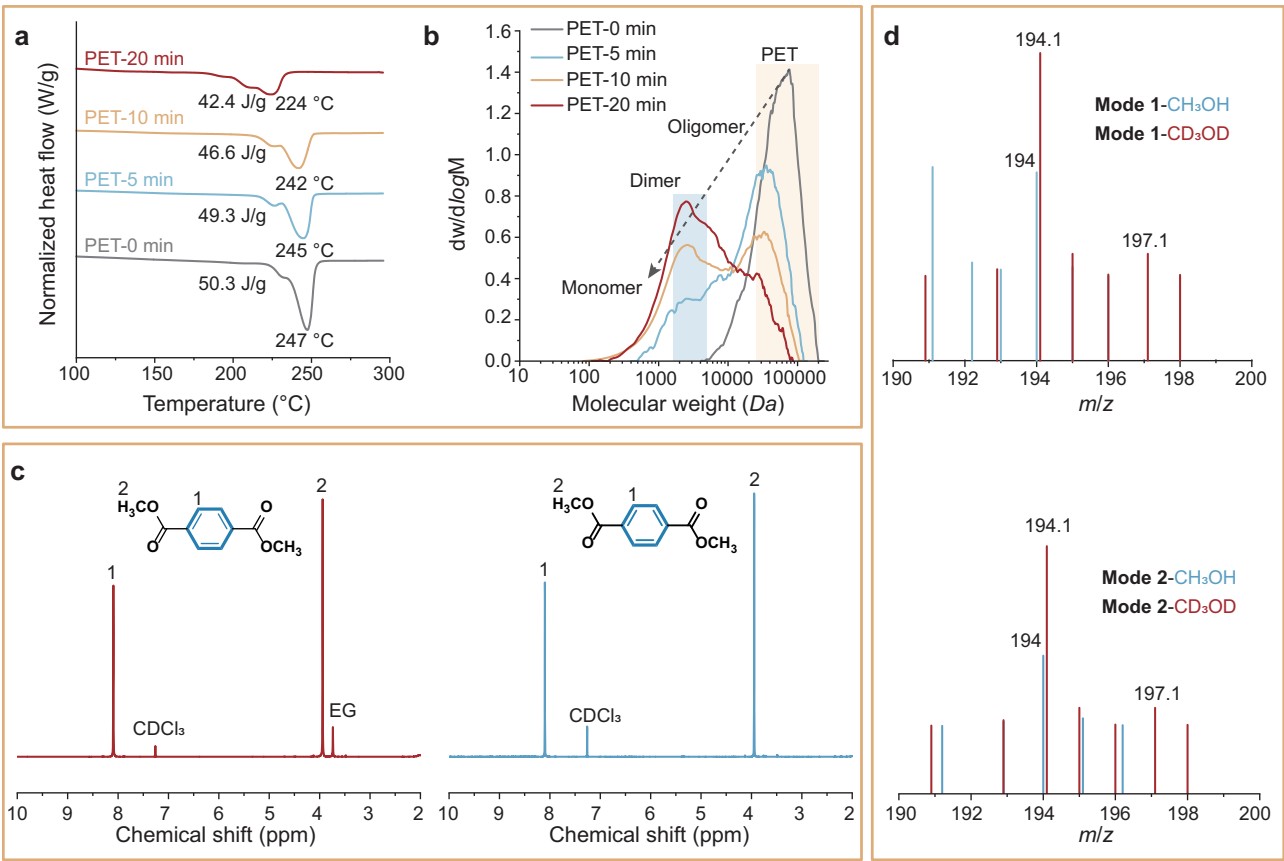

**Fig. 4 | Reaction pathways of PET depolymerization over $V_o$-rich Fe/ZnO NSs. a, b** DSC and GPC profiles of different reaction times over PET depolymerization. **c** $^1$H NMR spectrum of depolymerization products of **Modes 1** and **2** in DMSO-$d_6$. **d** Mass spectrogram of **Modes 1** and **2** depolymerization.

dominant role in promoting the crystalline partial fracture of polymer chains, converting the oligomers into DMT monomers[9].

In previous studies, the latter stage (i.e., the conversion of dimer into monomer) is considered a rate-limiting step for PET methanolysis, in which the bond breaking normally takes place at the C−O bond on the PET chain[49,50]. We synthesized two model dimers, i.e., 1,2-ethanediol dibenzoate (**Mode 1**) and deuterated $d_4$-1,2-ethanediol dibenzoate (**Mode 2**), using a method reported in previous studies (Supplementary Fig. 19)[51]. After depolymerization, the products of **Mode 1** comprised both DMT and 1,2-ethanediol (corresponding $^1$H NMR spectra in Fig. 4c). In contrast, the products of **Mode 2** only showed DMT monomers in $^1$H NMR spectra, where a peak for $d_4$-1,2-ethanediol was not observed. (Fig. 4c, and Supplementary Fig. 20).

We further performed isotope-labeling experiments were further conducted to investigate the action of methanol in the conversion of dimer into a monomer reaction. Synchrotron-radiation vacuum ultraviolet photoionization mass spectrometry (SVUV-PIMS) was employed to identify the reaction products during methanolysis of **Mode 1**, and **Mode 2** under real reaction conditions using methanol ($CH_3OH$) or methanol-$d_4$ ($CD_3OD$, 10 v% in $CH_3OH$) as solvents, respectively. Notably, $H_3CO(O)C-C_6H_4-C(O)OCD_3$ (corresponding to the signal at m/z = 197, Fig. 4d, Supplementary Fig. 21a, c) was derived from **Modes 1, 2** depolymerization. These products originated from the methanolysis of methanol containing 10 v% $CD_3OD$. Additionally, $H_3CO(O)$ $C-C_6H_4-C(O)OCH_3$ (corresponding to the signal at m/z = 194, Fig. 4d, Supplementary Fig. 21b, d) was produced from **Modes 1, 2** depolymerization of $CH_3OH$. These isotope labeling experiments provided strong evidence that the conversion **Modes 1** and **2** are derived from methanol. This finding verifies that **Modes 1** and **2** bond breaking position is at the $O=C-O-CH_2$ position of the dimer, not the

$O=C-O-CH_3$ position, thus leading to the formation of $H_3CO(O)$ $C-C_6H_4-C(O)OCD_3$ and $H_3CO(O)C-C_6H_4-C(O)OCH_3$.

To gain a deeper insight into the depolymerization mechanisms from polyester into DMT monomer, we employed in situ characterization and DFT calculations to monitor reaction intermediates. Experiments using in situ high-temperature-pressure infrared spectrometric (in situ HTP-IR) was carried out on a synchrotron source (Fig. 5a) revealing the depolymerization pathways by detecting intermediates and evaluating isotope effects during the reactions. As shown in Fig. 5b, depolymerization in **Mode 1** with $CH_3OH$ efficiently produces various intermediate species, including $v(C=O)$ at 1730 cm$^{-1}$, $v(C-O)$ at 1440 and 1259 cm$^{-1}$, $v(O-C-H)$ at 1338 cm$^{-1}$, $v(C-OH)$ at 1130 cm$^{-1}$ and $v(C-H)$ at 725 cm$^{-1}$ [52,53]. The hydroxyl group (1083 cm$^{-1}$) converts to triply bridged hydroxyl groups (1022, 1080, and 1118 cm$^{-1}$), indicative of the alkoxy bond formation between hydroxyl groups and the catalyst (Fig. 5b, Supplementary Fig. 22a). The intensity of $v(C-O)$ at 1440 cm$^{-1}$ and $v(O-C-H)$ at 1338 cm$^{-1}$ gradually increased, suggesting the activation of the hydroxyl group in $CH_3OH$[43,54]. Simultaneously, the peak intensities of these intermediates ($C=O^*$, $C-O^*$, $C-H^*$) gradually rise, indicating the gradual transformation of PET into DMT during the methanolysis process. In comparison, in situ FTIR spectra showed that the OD absorbance band at 2500 cm$^{-1}$ progressively increased and eventually stabilized with $CD_3OD$ adding (Fig. 5c, Supplementary Fig. 22b)[55]. As the reaction progresses, the OH bonds (3400 − 3700 cm$^{-1}$) on the surface are progressively increased to the generation of EG by **Mode 1** depolymerization (Supplementary Fig. 22a, b). In contrast, under the nitrogen atmosphere, the in situ IR spectrum of the depolymerization reaction does not exhibit the characteristic functional group changes mentioned above (Supplementary Fig. 23). Taken together, these vacancies can promote oxygen

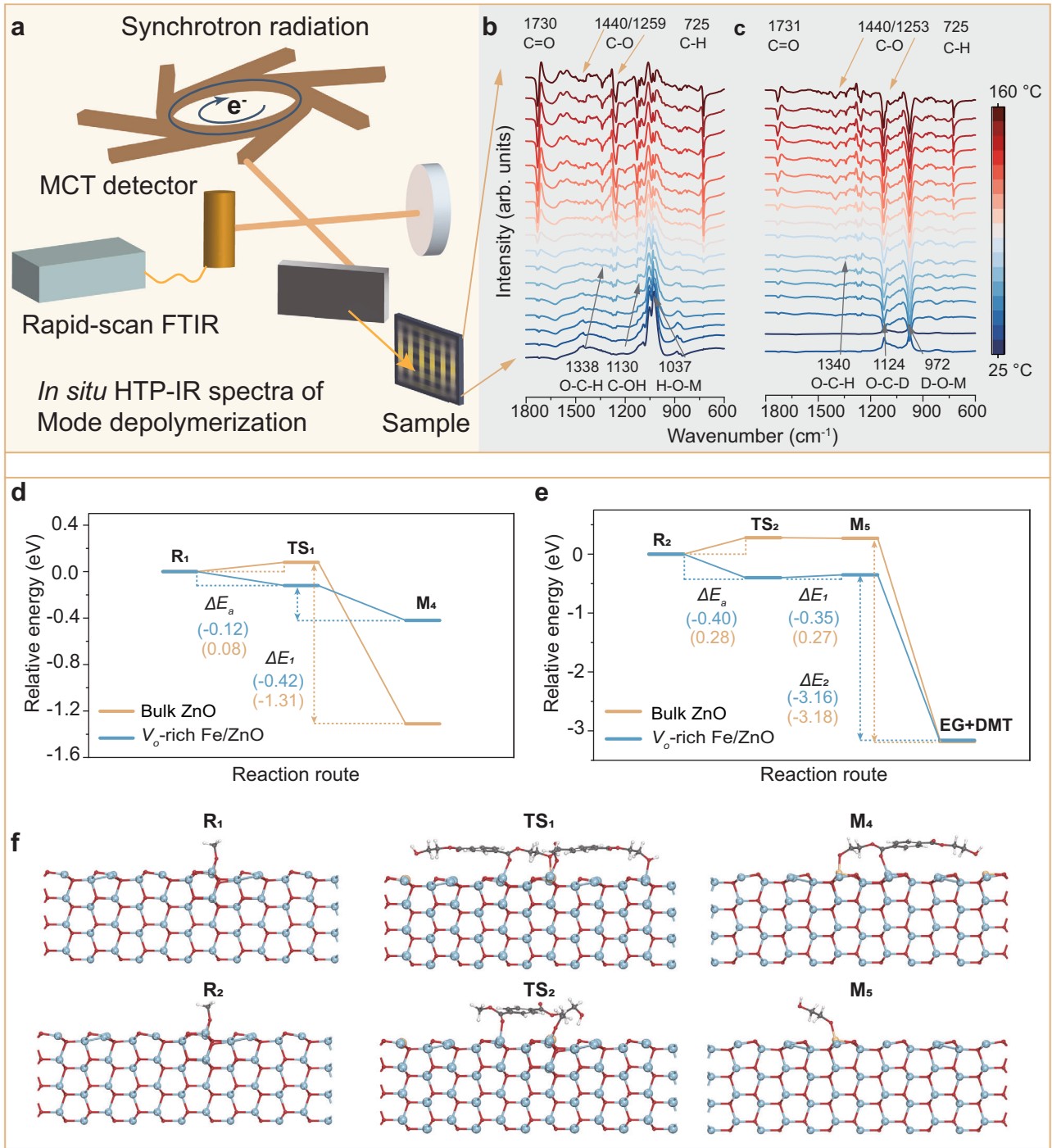

**Fig. 5 | Mechanism of PET depolymerization. a** Schematic illustration of in situ high-temperature-pressure infrared spectrometry spectra for **Modes 1** and **2** depolymerization processes under air atmosphere. **b, c** In situ high-temperature-pressure infrared spectrometry of **Mode 1** depolymerization in $CH_3OH$ and $CD_3OD$ under air atmosphere. **d–f** Free energy profiles for **Mode 1** on bulk ZnO, and $V_o$-rich Fe/ZnO NSs, with **f** showing the atomic structure change of **Mode 1** on the $V_o$-rich Fe/ZnO NSs.

dissociation and transfer, which are critical for the formation of the **M₃** structure. Thus, Zn and $CH_3OH^*$ on the **M₃** structure activate the carbon groups (O═C−O) of **Mode 1** and promote C−O bond breaking to form DMT and ethylene glycol.

To further understand the effect of oxygen vacancy in promoting the depolymerization of PET, we conducted DFT calculations for the depolymerization of PET on $V_o$-rich Fe/ZnO NSs and bulk ZnO. The initial structure **R₁** represents the **M₃** structure. Zn and $CH_3OH^*$ on the **M₃** structure activate the carbon group (O═C−O) of **Mode 1** to form an

intermediate species (**TS₁**). On the bulk ZnO surface, the formation of **TS₁** necessitates overcoming an activation energy of 0.08 eV (Fig. 5d, and Supplementary Fig. 24). Detailed data is provided in the SI (Supplementary Table 8). In contrast, on the $V_o$-rich Fe/ZnO NSs surface, the calculated barrier for the formation of **TS₁** decreases to −0.12 eV, substantially lower than that on bulk ZnO (Fig. 5d, f, Supplementary Fig. 25). Detailed data is also provided in the SI (Supplementary Table 9). These results underscore that the presence of the oxygen vacancy facilitates methanol and **Mode 1** adsorption on the catalyst

surface. This intermediate species is then adsorbed on the $M_3$ structure, where the C−O bond of **Mode 1** is broken, generating species $M_4$ product and monomeric DMT.

Similarly, the initial structure $R_2$ represents the $M_3$ structure. Zn and $CH_3OH^*$ on the $M_3$ structure activate the carbon group (O = C−O) of the 1-(2-hydroxyethyl) 4-methyl terephthalate (HEMT) product to form an intermediate species $TS_2$. On the bulk ZnO surface, the C−O bond cleavage of the HEMT necessitates surmounting the activation energy barrier ($TS_2$) of 0.27 eV (Fig. 5e). On the $V_o$-rich Fe/ZnO NSs surface, the $TS_2$ of −0.35 eV is lower than bulk ZnO (Fig. 5e, f). This result underscores that the formation of $CH_3OH^*$ species and $OOH^*$ species in the $V_o$-$Zn^{2+}$−O−$Fe^{3+}$ sites significantly enhances the cleavage ability of C−O, thereby accelerating ester bond activation and C−O bond cleavage. This species is then adsorbed on the $M_3$ structure, breaking the C−O bond of the $M_4$ product and thus generating ethylene glycol and DMT. Taken together, $V_o$-rich Fe/ZnO NSs promote the formation of the nucleophilic species $M_3$ structure from the adsorbed methanol and activate the carbon group of PET. This synergistic action catalyzes the breaking of the C−O in the ester bond, leading to the generation of DMT and ethylene glycol.

Based on the aforementioned results, Supplementary Fig. 26 provides a schematic representation of the methanolysis pathway of PET to DMT using $V_o$-rich Fe/ZnO NSs. The process begins with the $V_o$-$Zn^{2+}$−O−$Fe^{3+}$ adsorbing oxygen. These $O^{-*}$ species, in turn, attract the O−H bond from $CH_3OH$, forming $CH_3OH^*$ species and $OOH^*$ species. In the following step, the $CH_3OH^*$ species and $OOH^*$ species first attack the carbonyl carbon of PET, followed by the activation of the carbonyl oxygen of PET by $V_o$-$Zn^{2+}$−O−$Fe^{3+}$ metal sites. This synergistic effect enhances the cleavage of C−O bonds within PET, leading to the formation of oligomers. Over a series of processes involving multiple steps, including O−O bond and $CH_3OH$ activation, nucleophilic attack, C = O activation, and cleavage of the C−O bond (as mentioned above), the pristine PET is eventually converted into the end product of DMT monomer with high yield and high purity.

## Sustainability evaluation and life-cycle assessment (LCA)

Compared to high-grade pure PET waste (e.g., plastic bottles in Fig. 1c), recycling PET from textiles, carpets, and other waste materials, which consist of complex components, poses a greater challenge for catalytic depolymerization processes. To assess the efficiency of our approach, we first explored the methanolysis of polyester fibers and textiles containing minor additives like cellulose, adhesives, pigments, and crosslinkers. Despite the intricate composition potentially affecting the catalytic activity and hindering depolymerization, we successfully recovered 97-98% DMT from various PET waste, including felt, silk, and gauze (Supplementary Fig. 27, a1–a6). Our approach was further applied to polyester composites, including blends of PET with nylon 66 (5%), nylon 6 (20%), and acrylic (20%). Methanolysis of such waste resulted in 96-98% DMT recovery, leaving nylon or cellulose as residues (Supplementary Fig. 27, b1–b3). Additionally, tests on low-grade polyester materials like clothing, bags, and carpets consistently yielded over 95% DMT recovery (Supplementary Fig. 27, c1–c3). Taken together, these findings demonstrate that conventional additives and unknown impurities do not impede PET depolymerization, underscoring the efficacy of our method in processing complex PET waste. Notably, the pigment-laden DMT was effectively decolorized using activated carbon in hot methanol, yielding high-quality DMT with chromaticity (Hazen <10, Supplementary Fig. 27).

To evaluate the environmental impact, we proposed a conceptual model for a PET-waste recycling plant that integrates both recycling and conversion processes. This model was compared with traditional petroleum-based PET production. A life cycle assessment (LCA) was included 18 indicators of the entire process, which can be classified into three major categories including human health, ecosystems, and resources. LCA was conducted to mainly compare the global warming potential (GWP) and non-renewable energy use (NREU) of our methanolysis method with current industrial PET recycling methods (Fig. 6a and Supplementary Tables 10–12)[21,56]. The PET waste recycling technology boundary mainly included: (1) mechanical shredding of waste PET (collection, transportation, and pretreatment); (2) PET depolymerization (catalyst synthesis, PET methanolysis/separation); (3) re-polymerization; and (4) r-PET of extrusion. Detailed information on the LCA approach is provided in the SI (Supplementary Table 13, and Supplementary Figs. 28–29). The chemical recycling processes were simulated on an industrial scale with an annual treatment of 200,000 tons of waste PET, using Aspen Plus V11 to obtain the mass balance and energy consumption.

In terms of energy consumption, the production of virgin PET from petroleum consumes up to 90 MJ/kg in China and 70 MJ/kg in Europe (Fig. 6b and Supplementary Tables 14–19). Our methanolysis process reduced energy consumption by 56.0% and 40.9% in China (NREU = 46 MJ/kg) and Europe (NREU = 37 MJ/kg), respectively. The primary energy-intensive stages were mechanical shredding, depolymerization, and liquid separation, accounting for more than 64% of the total energy demand in both regions. Notable energy savings were achieved with the methanolysis of the PET recycling route.

Regarding carbon footprint, petroleum-based PET production resulted in 4.99 kg $CO_{2-eq}$/kg in China and 2.19 kg $CO_{2-eq}$/kg in Europe (Fig. 6c, Supplementary Tables 14–19). The PET depolymerization and re-polymerization stages were identified as the main contributors to greenhouse gas (GHG) emissions, primarily due to electricity and fuel consumption. In contrast, DMT production from the waste PET recycling route led to 2.76 kg $CO_{2-eq}$/kg in China and 2.15 kg $CO_{2-eq}$/kg in Europe. Consequently, recycled DMT from waste PET reduced total GHG emissions by 44.5% (China) and 1.8% (Europe) on a cradle-to-gate basis, demonstrating the potential of PET recycling for carbon neutrality and a lower carbon future. Taken together, the current case-CN or case-EU recycling route has less overall life cycle impact on human health, ecosystems, and resource categories than the virgin PET-CN or virgin PET-EU route (Supplementary Fig. 30).

A techno-economic analysis (TEA) was conducted to compare the system efficiency and cost of the conventional petroleum-based-terephthalic acid (TPA) route with our waste-PET-based DMT production[57]. Figure 6d, e illustrates the comparison of minimum selling prices (MSP) for TPA and DMT-production routes, with detailed cost breakdowns presented in Supplementary Tables 20–22. The high costs of sourcing clean PET (like the plastic bottles in Fig. 1) result in an MSP close to 1000 $/t, slightly lower than the price of traditional TPA routes (1021 $/t)[57,58]. In contrast, utilization of low-cost sourcing PET waste (like the mixed polyesters in Fig. 1) and PET textile waste (like the PET sources in Supplementary Fig. 27) led to 723 $/t and 425 $/t in the MSP, respectively. As such, the use of the cost-effective PET textile scrap, combined with $V_o$-rich Fe/ZnO NSs-assisted methanolysis, significantly lowers initial total operating costs by 58.4%.

## Discussion

In summary, our study has demonstrated the effectiveness of $V_o$-rich Fe/ZnO NSs in catalyzing the depolymerization of PET, achieving an exceptionally high STY of 502.2 $g_{DMT}\cdot g_{cat}^{-1}\cdot h^{-1}$ at 160 °C. This catalyst has displayed robust activity and stability in the depolymerization of various PET substrates, including PET, PET/PC, PET/PE, PET/PP, PET bottles, commercial textiles and fibers, mixed textiles with nylon, and degraded textiles. Furthermore, our research has contributed to a better understanding of the catalytic mechanisms involved in polyester depolymerization. Specifically, the $V_o$-$Zn^{2+}$−O−$Fe^{3+}$ sites on $V_o$-rich Fe/ZnO NSs were responsible for activating and breaking the O − O/O − H bond, leading to the formation of $OOH^*/CH_3OH^*$ species. Subsequent nucleophilic attacks and C = O activation of PET resulted in the gradual cleavage of the C−O bond, ultimately producing DMT. Moreover, LCA analysis indicated that this approach has the

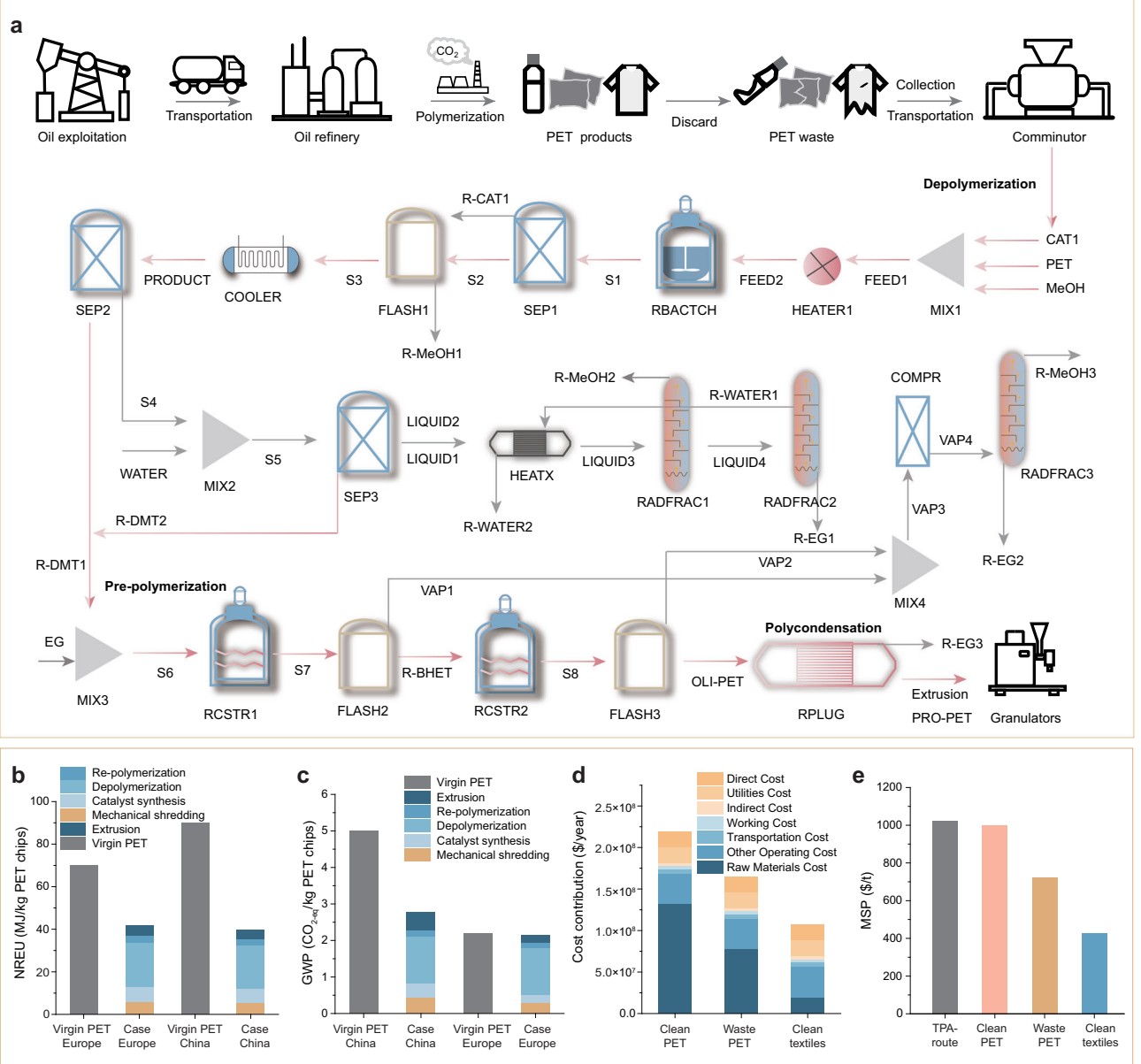

**Fig. 6 | Life-cycle assessment (LCA) and techno-economic analysis (TEA) of closed-loop PET recycling. a** Schematic diagram of the conventional petroleum-based PET production process, and newly-developed closed-loop PET recycling via the methanolysis route. Comparison of **b** non-renewable energy use (NREU) and **c** global warming potential (GWP). **d** Analysis of cost contribution based on clean PET, waste PET, and clean textiles recycling processes. **e** Comparison of minimum selling prices based on clean PET, waste PET, clean textiles, and the conventional petroleum-based TPA route.

potential to reduce the carbon footprint and enhance the energy efficiency of PET waste recycling. This methodology is suitable for converting various types of polyester, including low and high-quality PET, as well as mixed PET, into renewable DMT for use in the plastics industry. It offers a promising solution for the closed-loop recycling of PET.

## Methods

### Preparation of catalyst

Synthesis of oxygen-vacancy-controlled Fe/ZnO nanosheets. Zinc chloride anhydrous (40 mmol), ferric chloride (0.8 mmol), and L-alanine (4 mmol) were dissolved in 125 mL of ethanol and deionized (DI) water (v/v = 1/4) and vigorously stirred (800 rpm) until a homogeneous solution was formed. Ethanolamine (5 mL) was dissolved in 60 mL of DI water/ethanol (v/v = 1/2). Then, the mixed solution was added dropwise into the mixed solution under vigorous stirring

(800 rpm) at 25 °C for 6 h. The resulting products were washed with DI water and ethanol three times and dried under vacuum overnight. Oxygen vacancy-rich ultrathin Fe/ZnO nanosheets ($V_o$-rich-Fe/ZnO) were formed by calcining at 350 °C for 2 h in the air. Oxygen vacancy-poor ultrathin Fe/ZnO nanosheets ($V_o$-poor-Fe/ZnO) were formed by calcining at 500 °C for 2 h in the air.

Synthesis of bulk ZnO. Zinc chloride anhydrous (20 mmol) was dissolved in 125 mL of DI water to form a homogeneous solution. Sodium hydroxide solution (0.08 mol/L, 50 mL) was slowly dropped into the mixed solution under vigorous stirring (800 rpm) at 180 °C for 6 h. The resulting products were washed with DI water and ethanol three times and then dried in a vacuum overnight.

### Evaluation of catalytic performance

The depolymerization of the polyesters was carried out in a 50 mL stainless steel high-pressure autoclave with magnetic stirring. The

process is shown as follows: appropriate amounts of catalyst, 2.0 g PET, and 20 mL methanol were mixed in an autoclave. The polyester depolymerization reaction in the reactor is carried out under the air. To verify the effect of the reaction atmosphere on depolymerization performance, the polyester depolymerization reaction was purged with $N_2$ to eliminate residual air at ambient temperature. The reaction was conducted at 160 °C for 1 h with a stirring at 300 rpm. After the autoclave reaction, the liquid phase was separated from the catalyst by centrifugation and the quantitative analysis of liquid products was performed by a gas chromatography (Agilent 7820 A) equipped with an HP-5 capillary column and a flame ionization detector (FID). The catalyst activity was measured by testing the conversion of polyesters and the selectivity and yield of products, which were calculated using Eqs. (1), (2), (3) and (4). In depolymerization reactions, no polyesters remained after the reaction, indicating that the conversion was >99%. Moreover, the carbon balance was $100 \pm 3\%$, suggesting the selectivity generally equaled the yield.

$$\text{Conversion} = \frac{Polyester(input) - Polyester(residue)}{Polyester(input)} \times 100\% \quad (1)$$

$$\text{Selectivity}_i = \frac{n(product_i)}{\sum n(product)} \times 100\% \quad (2)$$

$$\text{Yield}_i = \frac{n(product_i)}{theoretical\ n(product)} \times 100\% \quad (3)$$

$$\text{Carbon balance} = \frac{\sum n(product_i)}{n(Polyester)} \times 100\% \quad (4)$$

### Characterization of catalyst and *r*-PET

Scanning electron microscope (SEM) images were obtained with a field emission microscope (JEOL-7100F). TEM and HRTEM images were collected with STEM/EDS microscopy (JEM-2100F, JEOL). High-angle annular dark-field scanning transmission electron microscopy (HAADF-STEM) image was collected with Themis Z scanning/transmission electron microscope. Atomic force microscope (AFM) images were collected with a MultiMode V system. The crystallographic characterization of the as-synthesized materials was obtained using an X-ray diffractometer (XRD) equipped with a Cu Ka X-ray source (D8 Advance, Bruker). Electron paramagnetic resonance (EPR) spectroscopy measurements were performed on a model spectrometer operating at the X-band frequency (JES-FA200, JEOL). The melting point of PET was analyzed by DSC (Discovery, TA Instruments) measurements. The test method was as follows: temperature from −90 to 160 °C (300 °C), and a heating rate of 10 °C/min under a nitrogen atmosphere. Molecular weight distributions of the pristine PET or the oligomer were analyzed on a gel permeation chromatography (Agilent PL-GPC 220), equipped with a PL-Gel Mixed B guard column, three PL-Gel Mixed B columns, and a refractive index (RI) detector. PET was dissolved in hexafluoroisopropanol, and the oligomer was dissolved in dichloromethane.

### Reporting summary

Further information on research design is available in the Nature Portfolio Reporting Summary linked to this article.

## Data availability

The data that supports the findings of the study are included in the main text and supplementary information files. Raw data can be obtained from the corresponding author upon request. Source data are provided with this paper.

## Code availability

The code that supports the findings of this study is available from the corresponding author upon request.

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

## Acknowledgements

We gratefully acknowledge the financial support from the National Science Foundation of China (52170094 to X.Lu, 52303140 to J.C.) and the Fundamental Research Funds for the Central Universities to X.Lu. This study also received partial support from the National Key R&D Program of China (2023YFC3903200 to J.C.) and the Postdoctoral Fellowship Program of China Postdoctoral Science Foundation (GZB20230696 to J.C.). Synchrotron-radiation SVUV-PIMS experiments and IR measurements were respectively conducted at the Beamline BL04B (2022-HLS-PT-005837) and Beamline BL01B (2022-HLS-PT-004818) of the National Synchrotron Radiation Laboratory (NSRL). In situ IR experiments were performed on Beamline BL06B1 (2023-SSRF-PT-503071, 2023-SSRF-JJ-503215, 2023-SSRF-JJ-503307) at the Shanghai Synchrotron Radiation Facility (SSRF). The computational analysis involving DFT calculations was conducted at the Supercomputing Center of Max Planck Computing & Data Facility (MPCDF) and the

Supercomputing Center at the University of Science and Technology of China (USTC). Some material characterization for this study was carried out at the USTC Center for Micro- and Nanoscale Research and Fabrication, and the Instruments Center for Physical Science at USTC. The authors also thank Mr. Huijie Duan for assistance in creating schematic illustrations.

## Author contributions

J.C. and X.Lu conceived the idea and designed experiments. J.C., H.L., and Z.Z. performed experiments. H.L. conducted LCA analysis, with support and assistance received from J.Y. and J.D.; X.Li contributed to the DFT calculations and analysis. J.C., H.L., M.E., and X.Lu contributed to the data interpretation. J.C., M.E., and X.Lu co-wrote the paper. All authors discussed the results.

## Competing interests

The authors declare no competing interests.
