## [Peer Review File · Nature Communications]

Depolymerization mechanisms and closed-loop assessment in polyester waste recyclingReviewers' comments:

Reviewer #1 (Remarks to the Author):

The paper is focused on the degradation of PET via heterogeneous catalyst. Detailed LCA and characterisation of the mechanism have been undertaken. The authors look at mixed waste - I do feel that better reference to the literature should be given here, as this is known to be affected by temperature. I am a little surprised there is not mention of mass transport effects - you have the huge polymer trying to get close to the catalyst - I wonder if the authors have thought of any effect from this?

It must be stated that impressive efforts have been undertaken to characterise the system and the mechanistic understanding is interesting and important to the field. I would recommend acceptance.

Reviewer #2 (Remarks to the Author):

Based on the present form, I feel that this manuscript is critically flawed in several aspects, which are specified as follows:

1. In the abstract, the performance and conditions of catalyzing can be compared.
2. The novelty of this approach is insufficiently explained. The pros and cons of related approaches are well-known, thus the key issues that the manuscript intends to address need to be highlighted in the early stage of the introduction sector.
In addition, the focus on alcoholysis also requires proper explanation, since the authors claimed that "Hence, there is a crucial need to develop a green, efficient, and sustainable alcoholysis method for converting real PET wastes into monomer units".
3. Please define "some promising results from lab-scale experiments".
4. The conceptual coherence of "alcoholysis" and "methanolysis" shall be explained.
5. The scientific significance of the summarized contributions is rather insufficient. The discussion on "First, PET methanolysis pathways and mechanisms are poorly understood, particularly regarding the interactions between solvent, catalyst, and polymer" can be enriched to underscore the science of this work.
6. Focusing on only two indicators, this life cycle assessment can never be regarded as "a comprehensive life cycle assessment for the entire process".
7. The authors need to ensure the same depolymerization conditions, instead of "similar depolymerization conditions".
8. If the authors intended to uncover "the interactions between solvent, catalyst, and polymer", orthogonal experimental design can be considered in this work.
9. The LCA procedure lacks basic elements (e.g., four basic steps) and sufficient life cycle inventory. Therefore, the LCA analysis seems incomplete and unconvincing.
10. Detailed processing parameters can be used to replace vague words. For instance, in Page 16, the authors need to specify the number of rpm, instead of using "vigorous stirring". Further, I recommend that references can be added to support the selection of processing parameters.

Reviewer #3 (Remarks to the Author):

The authors synthesize a PET alcoholysis catalyst consisting of nanosheets of ZnO and Fe. They tested a wide range of waste PET and plastic mixtures and showed the catalyst gave very high conversions and yields when impurities were present. The DMT monomer could successfully be repolymerized to PET showing the possibility of closed-loop recycling. They characterized the catalyst, then performed DFT-informed in situ IR experiments, and deuterium-labeling studies. An LCA was included showing the process could improve the sustainability relative to virgin PET production.

This is a topic of much interest recently and the range of materials tested, and the effectiveness of the catalyst are both impressive. It is also appropriate to include many of the characterization techniques used and the inclusion of an LCA is also appropriate and useful. However, the purpose of the experiments are in general not well explained, there are numerous inconsistencies, and many important details are not included. Therefore, I recommend this manuscript should be rejected, although there are many suggestions included below the authors may consider to improve this piece of research.

Line 35-36. Specify that PET can be recycled into BHET or DMT. The end product depends on the reaction.

Lines 65-68. Most alcoholysis operates with a metal salt catalyst, not with strong acids or bases or under supercritical conditions. The description provided could be misleading and should be changed to reflect the general practice of the field.

Lines 77-83. The listed needs do not directly lead to the conclusion of needing new catalysts. The list of challenges should be amended to include higher activity catalysts.

Line 84. "NS" needs to be defined as nanosheets before using the abbreviation.

Lines 130-131. Grammar

Lines 133-134. Why not run the PET/PC reaction at 160C and depolymerize everything simultaneously? Mechanical sorting is cited as the issue being addressed here but surely the stepwise reaction is meant to allow for distillation of BPA from unreacted PET to avoid chemical separations of the mixture of products, not mechanical separations. Please clarify.

Line 147. The authors should clarify what is meant when the catalyst possesses structural variables.

Line 150. The catalyst regeneration process used for catalyst recyclability tests should be noted somewhere. I do not see it anywhere.

In Figure 2a, the two depicted oxygen vacancies are not drawn the same. There is a 2-coordinate Zn near the left vacancy while the same location has a 3-coordinate Zn near the right vacancy.

Lines 192-193. The authors make a comparison the "other electronic slabs." This is in reference to the difference between a surface layer vs the second layer deep. It is suggested to simplify the language and say, "the second layer" instead of "other electronic slabs."

Lines 193-194. The presence of five slabs of Fe/ZnO gave enhanced vacancy generation. Please state what this was compared to.

Line 201. The designation of iron in the $Zn^{2+}-VO-Fe^{\delta+}$ with a delta should be explained.

Lines 203-204. It is known that ESR sensitivity is inversely correlated to temperature, intrinsically. It cannot be said that the oxygen vacancy density changes when temperature changes without more careful standardization at the compared temperatures.

Lines 205-207. The author appears to be attributing the difference with oxygen vacancy concentration at higher temperatures to gradual adsorption of oxygen. How much of the actual signal loss is attributable to this effect vs a temperature-dependent oxygen vacancy stability vs some intrinsic ESR temperature-dependence? This data needs more clear explanation of its importance.

Lines 222-224. The claim that triply bridged hydroxyl groups indicates the formation of alkoxy bonds between hydroxyl groups and Fe-Zn is not obvious and deserves more explanation.

Lines 227-229. This statement implies that Fe^{3+} donates d electrons to the bridging oxygen. It should be rewritten to clarify that electron transfer is from oxygen to Fe. Figure 3d also implies with the arrow direction that there is d-electron donation from Fe to O while the second statement in this sentence implies that there is electron donation from O to Fe.

Figure 3d shows four electrons in a single Zn d-orbital which, of course, cannot be the case.

It is necessary to clarify what is meant by an oxygen vacancy. If it is meant that the Zn-VO-Fe site of interest is actually a Zn-O-Fe sites adjacent to the oxygen vacancy, as is suggested in Figure 3d, this must be made extremely clear to avoid the common confusion surrounding oxygen vacancies.

Lines 230-231. It is not made clear how the claim that the proposed site favored the adsorption of the nucleophilic group is supported. Please state how this is known.

Lines 232-233. The authors claim that O_2 activation is an important reaction for methoxy formation. However, the reactions are run under a N_2 atmosphere, so it is confusing that O_2 is invoked. An alternative route should be explored that does not rely on O_2 .

Line 243. The authors state that methanol is dehydrogenated when it is only deprotonated. Methanol dehydrogenation would lead to formaldehyde, not methoxy species.

Lines 260 and 270. Colored shaded boxes are referred to in Figure 4, but these are not present. Figure 4c. This figure shows a branched polyethylene polymer it appears. It should reflect the structure of PET better by being linear.

The entirety of the isotopic labeling section is lacking in explanation and context. As it is currently, it is difficult to gain any insights from the experiments.

Figure 4d. The structures in the middle panel have an extra carbon in the esters.

Lines 299-300. The authors should be careful in their use of the terms radicals and deprotonation. Deprotonation is a heterolytic process whereas radical generation would be homolytic in this case. This terminology should be updated to reflect the chemistry more accurately.

Lines 453-470. The source and (particle) size of PET used for optimization reactions needs to be specified. Additionally, the HPLC (for glycolysis reactions) and GC (for methanolysis reactions) methods should be added to the SI with an example chromatograph.

Depolymerization reactions were carried out after purging of air with nitrogen, but in situ IR experiments were carried out under air. Since O₂ is thought to be an active species, this is an important difference and in situ experiments should be run under nitrogen. There should be ICP reported for the catalyst synthesis. The provided TEM image does not clearly show vacancies, especially not 10% of the surface metal as would be suggested by the ratio of Zn to Fe in the synthesis without further characterization.

REVIEWER #1

Before our response to the specific comments of the reviewer, we would like to highlight the key revisions made:

1. Revising the Introduction section to highlight the challenges of existing PET recycling strategies and emphasize the novelty and scientific contribution of our study.
2. Adding detailed descriptions of our experimental procedures, including depolymerization processes, catalyst regeneration procedures, and chromatography testing conditions.
3. Conducting *in situ* FTIR measurements of PET depolymerization under a nitrogen atmosphere, providing insights into the role of oxygen species in the catalyst-mediated activation of methanol and depolymerization.
4. Performing *in situ* EPR measurements of PET depolymerization at various reaction temperatures and atmospheres (Figure 3b).
5. Adding a new figure (Figure S30) that includes all 18 indicators evaluated in our comprehensive life cycle assessment of PET depolymerization processes.

General Comment: *The paper is focused on the degradation of PET via heterogeneous catalyst. Detailed LCA and characterisation of the mechanism have been undertaken. The authors look at mixed waste - I do feel that better reference to the literature should be given here, as this is known to be affected by temperature. I am a little surprised there is not mention of mass transport effects - you have the huge polymer trying to get close to the catalyst - I wonder if the authors have thought of any effect from this? It must be stated that impressive efforts have been undertaken to characterise the system and the mechanistic understanding is interesting and important to the field. I would recommend acceptance.*

Our Response: We sincerely appreciate the reviewer's compliments regarding the the novelty and significance of our work. A detailed response to the reviewer's comments is presented below.

Comment 1: *The authors look at mixed waste - I do feel that better reference to the literature should be given here, as this is known to be affected by temperature.*

Our Response: We would like to thank the reviewers for their constructive comments. We now include references to recent studies that explore the chemical recycling processes of mixed plastics. For example, Saito et.al. developed an organocatalyst that selectively deconstructs condensation polymers at specific temperatures. Such a process allows for the easy separation of additives, or other polymers, such as polyolefin or cellulose, from mixed plastics, thereby providing a viable chemical recycling pathway for existing mixed plastics ¹(*Mater. Horiz.*, 2023,10, 3360-3368). Additionally, Li et.al. reported a novel approach using pioneering single-atom catalysts to convert mixed plastic wastes into a uniform chemical product, where the single-atom Ru catalyst achieves approximately 90% conversion of real mixed plastic

wastes into methane with a selectivity exceeding 99%² (*J. Am. Chem. Soc.* 2023, 145, 41, 22836–22844). Following the reviewer’s suggestion, we have added these references to clarify the treatment and analysis of mixed plastic waste in our study:

Added reference to mixed plastic waste depolymerization section (Lines 152-155) manuscript:

“Such a result demonstrates the utilization of temperature differences for selective and sequential depolymerization of polyester mixture wastes (like PC/PET) using V_o -rich Fe/ZnO NSs catalyst³⁸⁻⁴⁰. ~~Given the challenging nature of the mechanical separation of mixed polyester plastic wastes, this could present a useful strategy for recovering valuable products. This approach enables the selective chemical depolymerization of PET/PC mixed plastics to yield monomers, i.e., DMT (from PET) and BPA (from PC), effectively overcoming the challenges associated with mechanical sorting and separating mixed depolymerization products.”~~

Comment 2: *I am a little surprised there is not mention of mass transport effects - you have the huge polymer trying to get close to the catalyst - I wonder if the authors have thought of any effect from this?*

Our Response: The reviewer raised an important point for our consideration and we also agree on the significance of mass transfer effects between catalysts and polymers. In the preliminary stage, we performed comparative experiments and the results show that the size of the polymer indeed affects the catalytic mass transfer efficiency. As shown in the figure below (Figure R1), when the particle size of polyester particles is ~5 mm, the space-time yield is 66% relative to the yield using the particle size of ~1 mm. We have also investigated the impact of polyester molecular weight on catalytic performance. The results demonstrated that the space-time yield of low molecular weight PET (pure bottles $M_w=38.2$ kDa, or textiles $M_w=23.8$ kDa) is significantly higher than the space-time yield of the polyester particles ($M_w=59.1$ kDa). Overall, our data indicate that the polymer's size and molecular weight are critical in determining mass transfer efficiency, thus governing the depolymerization efficiency.

Figure R1. Comparison of the space-time yields of V_o -rich-Fe/ZnO with different particle sizes and molecular weights of PET.

REVIEWER #2

Before our response to the specific comments of the reviewer, we would like to highlight the key revisions made:

1. Revising the Introduction section to highlight the challenges of existing PET recycling strategies and emphasize the novelty and scientific contribution of our study.
2. Adding detailed descriptions of our experimental procedures, including depolymerization processes, catalyst regeneration procedures, and chromatography testing conditions.
3. Conducting *in situ* FTIR measurements of PET depolymerization under a nitrogen atmosphere, providing insights into the role of oxygen species in the catalyst-mediated activation of methanol and depolymerization.
4. Performing *in situ* EPR measurements of PET depolymerization at various reaction temperatures and atmospheres (Figure 3b).
5. Adding a new figure (Figure S30) that includes all 18 indicators evaluated in our comprehensive life cycle assessment of PET depolymerization processes.

General Comment: *Based on the present form, I feel that this manuscript is critically flawed in several aspects, which are specified as follows:*

Our Response: We are grateful to the reviewer for their insightful comments and efforts in improving our study. We have carefully considered each suggestion and comment to enhance the quality of the manuscript. In this revised version, we highlight our innovative use of the oxygen-vacancy (V_o)-rich Fe/ZnO nanosheets for catalyzing the recycling of polyester waste. This approach not only minimizes environmental impacts but also promotes resource sustainability.

Moreover, in response to the reviewer's major comments, we would like to highlight the following points. First, our study provides in-depth elucidation of the reaction mechanisms involved in polyester depolymerization by applying *in situ* characterization and density functional theory (DFT) calculation. Second, our analysis shows that the use of a small amount of V_o -rich Fe/ZnO catalyst in PET depolymerization demonstrates the viability of our approach in closed-loop recycling. Third, LCA analysis reveals that such a catalytic process achieves substantial energy savings and a significant reduction in greenhouse gas emissions. The analysis considered all potential impacts of the closed-loop process, including the effects on human health, ecosystems, and resources across 18 evaluated categories.

Detailed responses to the reviewer's comments are as follows:

Comment 1: *In the abstract, the performance and conditions of catalyzing can be compared.*

Our Response: Following the reviewer's suggestion, we have now included a comparative analysis of the space-time yield (STY) results of methanolysis and glycolysis using our oxygen-rich vacancy catalysts in air or N₂ atmospheres. This analysis revealed significant PET depolymerization performance enhancement in air. In particular, under methanolysis and glycolysis in the air compared to N₂, the STY for dimethyl terephthalate (DMT) and bis(2-hydroxyethyl) terephthalate (BHET) improved by 50.5-fold and 28.2-fold, respectively.

To clarify these findings in the manuscript, we have revised the Abstract as follows:

Line 37-40: *“Here, we introduce an efficient PET-alcoholysis approach utilizing an oxygen-vacancy (V_o)-rich catalyst under air, achieving space-time yield (STY) of $505.2 \text{ g}_{\text{DMT}} \cdot \text{g}_{\text{cat}}^{-1} \cdot \text{h}^{-1}$ and $957.1 \text{ g}_{\text{BHET}} \cdot \text{g}_{\text{cat}}^{-1} \cdot \text{h}^{-1}$, these results represent 51-fold and 28-fold performance enhancements compared to reactions conducted under N₂.”*

Comment 2: *The novelty of this approach is insufficiently explained. The pros and cons of related approaches are well-known, thus the key issues that the manuscript intends to address need to be highlighted in the early stage of the introduction sector. In addition, the focus on alcoholysis also requires proper explanation, since the authors claimed that “Hence, there is a crucial need to develop a green, efficient, and sustainable alcoholysis method for converting real PET wastes into monomer units”.*

Our Response: We thank the reviewer for the comments. In response to the reviewer's suggestion, we have revised the **Introduction** to more explicitly highlight the challenges and necessity of developing a sustainable strategy for the alcoholysis of polyester.

Alcoholysis involves using alcohol reagents to depolymerize PET into corresponding monomers. A key aspect of this process is the catalyst design. Existing processes reported in the literature typically use homogeneous metal salts, ionic liquids, enzymes, or other catalysts ³(*Chem. Rev.* 2024, 124, 5, 2617–2650). While these catalysts achieve adequate efficiency, their depolymerization processes pose challenges such as difficulty separating the product from the catalyst, metal ion leaching, and poor catalyst stability. These challenges thus necessitate the development of a green, efficient, and sustainable alcoholysis catalyst for converting real PET waste into monomers.

A recent study by Ding et al. demonstrated a strategy of constructing a solid-solid interface between plastic and catalyst, enabling methanolysis through methanol vapor. Such interfacial catalysis could pave new pathways for plastic waste depolymerization. Moreover, other efforts have shown that oxygen vacancy-metal oxide catalysts can effectively catalyze the depolymerization of PET to generate BHET monomers ^{4,5}(*ACS Sustainable Chem. Eng.* 10, 5476-5488, (2022); *iScience* 26, 107492 (2023)). For example, Wang et al. demonstrated the use of KH550-modified defect-rich CeO₂ nanoparticles in glycolysis PET, achieving a conversion rate of 98.6%, and a BHET yield of 90.3% at the critical boiling point of ethylene glycol (196 °C) ⁶(*ACS Sustainable Chem. Eng.* 2022, 10, 5278–5287). Despite such adequate

progress, the application of these catalysts to real polyester waste (such as cotton blends, textiles, and undegraded carpets) has not been well validated, thus remaining substantial technical challenges.

In response to the reviewer's suggestion, we have revised the Introduction to more explicitly highlight the challenges and necessity of developing a sustainable strategy for the alcoholysis of polyester. The revised text now reads:

Line 63-79: ~~“Among these chemical methods, one of the promising routes towards achieving industrial closed-loop recycling of real PET wastes is alcoholysis, namely methanolysis and glycolysis. However, this alcoholysis largely operates under harsh conditions, including the use of caustic strong bases and acids, which cause severe equipment corrosion and generate hazardous liquid waste²⁸, or under high temperatures (e.g., >200 °C)^{29,30}. Such challenges limit the real applications of these chemical recycling methods in PET waste management. Hence, there is a crucial need to develop a green, efficient, and sustainable alcoholysis method for converting real PET wastes into monomer units. Alcoholysis is a process that utilizes alcohol as a solvent to depolymerize PET into monomers. The efficacy of this process largely relies on the design of the catalyst. Previous studies on this process typically involved the use of catalysts like homogeneous metal salts, ionic liquids, enzymes, and other catalysts²⁸⁻³⁰. While these catalysts achieve adequate efficiency, challenges such as difficulty in separating the product from the catalyst, leaching of metal ions, or catalyst instability remain prevalent. Consequently, developing a green, efficient, and sustainable catalyst for alcoholysis to convert PET into monomers is imperative.~~

~~Recent studies have shown that oxygen vacancy metal oxide catalysts can effectively catalyze depolymerization of PET to yield bis-2-hydroxyethyl terephthalate (BHET) monomer through glycolysis³¹⁻³³, or dimethyl terephthalate (DMT) through methanolysis. While these studies have reported some promising results from lab-scale experiments, scaling up the developed methods into real applications is hindered by the following challenges³⁴⁻³⁶. A recent study introduced the concept of creating a solid-solid interface between plastic and catalyst, allowing methanolysis of the polymer through methanol vapor²⁶. This innovative approach to interfacial catalysis opens up new avenues for the depolymerization of plastic waste. Additionally, other studies also demonstrate the promise of using oxygen vacancy-metal oxide in catalyzing the depolymerization of PET to produce BHET monomers³¹⁻³³. For instance, defect-rich CeO₂ nanoparticles modified with KH550 have been used for the glycolysis of PET. , at the critical boiling point of ethylene glycol (196 °C), a PET conversion rate of 98.6% with a BHET yield of 90.3% was achieved using this approach³⁴.”~~

Comment 3: Please define “some promising results from lab-scale experiments.”

Our Response: We appreciate the reviewer's effort in improving our manuscript. The phrase “some promising results from lab-scale experiments” has been revised for clarification. The revised text now reads:

Lines 80-82: “... ~~Some promising results from lab scale experiments.~~ Despite these encouraging findings, there are limited efforts on validating catalysts for real polyester waste, such as cotton blends, textiles, and undegraded carpets, leaving several challenges unaddressed³⁵⁻³⁷.....”

Comment 4: *The conceptual coherence of “alcoholysis” and “methanolysis” shall be explained.*

Our Response: We thank the reviewer for highlighting the need for clearer explanations regarding the term “alcoholysis” and “methanolysis”.

In particular, alcoholysis refers to the process of using various alcohols as solvents—such as methanol, ethanol, ethylene glycol, and octanol—to facilitate the depolymerization of PET into its corresponding monomers. These monomers include dimethyl terephthalate (DMT) from methanol, diethyl terephthalate (DTA) from ethanol, bis(2-hydroxyethyl) terephthalate (BHET) from ethylene glycol, and dioctyl terephthalate (DOPT) from octanol, through the breaking of chemical bonds.

Methanolysis is a specific type of alcoholysis, that exclusively uses methanol as the solvent to promote the depolymerization of PET into DMT. This distinction is crucial as it impacts the choice of catalyst and the efficiency of the depolymerization process under different conditions.

Following the reviewer’s comment, we added the following text to the Introduction. The revised text now reads:

Lines 34-37: *Alcoholysis of poly(ethylene terephthalate) (PET) waste to produce monomers, including methanolysis to produce dimethyl terephthalate (DMT) and glycolysis to produce bis-2-hydroxyethyl terephthalate (BHET), is a promising strategy in PET waste management.*

Comment 5: *The scientific significance of the summarized contributions is rather insufficient. The discussion on “First, PET methanolysis pathways and mechanisms are poorly understood, particularly regarding the interactions between solvent, catalyst, and polymer” can be enriched to underscore the science of this work.*

Our Response: With the reviewer’s suggestion, we realize the need to articulate the significance and contribution of our study, especially regarding the mechanisms of PET depolymerization. In response, we have revised the Introduction to incorporate a more detailed discussion of existing challenges of the alcoholysis process.

In particular, we now provide a detailed exploration of the interaction mechanisms at the interface of the solid phase (catalyst), solvent (methanol), and solid phase (polymer) during PET depolymerization. We also discuss the catalyst’s effectiveness on actual plastics and the lifecycle assessment (LCA) of the PET depolymerization process. The major points include:
(1) Unclear Depolymerization Pathways and Mechanisms: Existing literature largely

focuses on catalyst design and depolymerization efficiency, while comprehensive studies on PET depolymerization pathways and mechanisms are scarce. Such knowledge gap is primarily attributed to the intricate nature of the alcoholysis process, which involves sophisticated high-temperature, solid-liquid-solid interface reactions involving the catalyst, PET polymer, and solvents. **(2) Challenges in Recycling Real Mixed-PET Plastics:** Real PET wastes like plastics and textiles often contain additives such as pigments and plasticizers. These additives likely interfere with catalytic efficiency. However, existing progress in the literature mostly demonstrates a proof-of-concept using pure, clean PET. There is a lack of studies evaluating the recycling efficiency of real PET plastics and textiles. **(3) Lack of Life Cycle and Economic Analysis:** Despite the potential of catalytic depolymerization processes for sustainable management of PET plastic wastes, comprehensive analysis of the life cycle, economic, and technical aspects of the recycling process is relatively scarce. Such a fact largely limited the systematic evaluation of system efficiency and process viability.

With the reviewer's comments, we have included these points in the **Introduction** to clarify the scientific foundation and significance of our study:

Lines 82-94: *~~“First, PET methanolysis pathways and mechanisms are poorly understood, particularly regarding the interactions between solvent, catalyst, and polymer. Second, efforts on recycling real PET plastics, as well as validating the effectiveness of monomer reuse, are still limited. Third, a comprehensive life cycle assessment for the entire process of PET alcoholysis is also lacking. First, most articles focus solely on catalyst design and PET depolymerization efficiency, with few investigating the pathways and mechanisms of PET depolymerization. This knowledge gap exists because the alcoholysis process involves high-temperature, solid-liquid-solid complex interface reactions among catalysts, polymers, and solvents. Moreover, there is a lack of research on the interactions between catalysts and solvents, solvents and polymers, and catalysts and polymers during the reaction process. Second, few studies examine the recycling of actual PET plastics or textiles, which contain additives like pigments and plasticizers that can affect the catalytic efficiency. Third, while PET recycling aims to recover waste, comprehensive life cycle evaluations, as well as economic and technical analysis of the entire recycling process, are scarce. Therefore, innovative catalyst design and deeper insights into reaction mechanisms, are critically needed for closed-loop recycling of real PET waste.”~~*

Comment 6: *Focusing on only two indicators, this life cycle assessment can never be regarded as “a comprehensive life cycle assessment for the entire process”.*

Our Response: We would like to thank the reviewer for the insightful comments. Actually, we did perform a comprehensive LCA that included 18 indicators of the entire process (Figure R2), which can be classified into three major categories including human health, ecosystems, and resources. In our initial manuscript, we only included the results from two primary indicators—**carbon emissions** (Figure 6b) and **non-renewable resource consumption**

(Figure 6c)—that are most widely realized in the literature. Notably, such extensive assessment demonstrates that the overall life-cycle impacts of the polyester recycling routes are significantly lower in terms of human health, ecosystems, and resource use compared to those from the production of virgin PET. This finding further strengthens our claim on the promise and potential of the developed catalytic process for sustainable PET waste management.

Figure R2. Comparative life-cycle assessment results of case-CN or case-EU recycling route and virgin PET-CN or virgin PET-EU route.

Following the reviewer’s comments, we have decided to add the results of other indicators to the manuscript and add relevant text to the manuscript.

Line 460-463: *“Taken together, the current case-CN or case-EU recycling route has less overall life cycle impact on human health, ecosystems, and resource categories than the virgin PET-CN or virgin PET-EU route (Supplementary Figure 30).”*

Added a new Figure S30 to the Supporting Information:

Supplementary Fig. 30. Comparative LCA results of case-CN or case-EU recycling route and virgin PET-CN or virgin PET-EU route.

Comment 7: *The authors need to ensure the same depolymerization conditions, instead of “similar depolymerization conditions”.*

Our Response: We thank the reviewer for the valuable comment. We recognize the importance of maintaining consistency in depolymerization conditions when comparing results across different studies. The original term “*similar depolymerization conditions*” could indeed lead to confusion among readers by suggesting a variability that might affect the comparability of results. For clarification, we have made the following revisions:

In Lines 118-141, we now differentiate the depolymerization conditions used in various studies to ensure clarity and accuracy in our comparisons.

In Figure 1a, we made revisions to differentiate three specific conditions for glycolysis: (1) at the same temperature; (2) at the critical boiling point of ethylene glycol; and (3) at the melting point temperature of PET. These modifications ensure that our comparative analysis is precise, reducing any potential misunderstandings related to the experimental conditions.

In Figure 1b, we have classified the methanolysis conditions into four distinct categories: (1) mixed solvents (tetrahydrofuran/chloroform) assisted methanolysis; (2) homogeneous catalytic methanolysis; (3) heterogeneous methanolysis; (4) heterogeneous layer-assisted methanolysis. The heterogeneous layer-assisted process illustrates the space-time yield ($g_{\text{PET}} g_{\text{cat}}^{-1} h^{-1}$).

To address the reviewer’s concerns, we made revisions to Figures 1a-b and edited the corresponding description for clarification.

Lines 113-131: ~~*“The performance of the synthesized catalyst (V_o -rich Fe/ZnO NSs) was evaluated and compared with that of reported catalysts used under similar depolymerization conditions, as summarized in Figure 1a. Our experiments conducted the glycolysis process at 180 °C for 1 h under air. PET was completely converted to BHET (approaching 95.5% yield), which was detected by high-performance liquid chromatography (HPLC). Our V_o -rich Fe/ZnO NSs catalyst could convert PET into BHET with much higher activity ($957.1 g_{\text{BHET}} g_{\text{cat}}^{-1} h^{-1}$) than the catalysts in the literature (Figure 1a left panel and Supplementary Table 1). Considering different reaction conditions used in previous studies, we divided the performance data of glycolysis into three categories (Figure 1a): (1) glycolysis at the same temperature; (2) glycolysis at the critical boiling point of ethylene glycol; and (3) glycolysis at the melting point temperature of PET. Similarly, PET flake wastes could be completely converted through methanolysis in 1 h at 160 °C to obtain DMT with 99% yield (>99.5% purity) detected by gas chromatography. In particular, the presence of V_o -rich Fe/ZnO NSs catalyst results in a DMT formation rate of $505.2 g g^{-1} h^{-1}$, which is an order of magnitude higher than that of the reported*~~

catalysts for methanolysis (Figure 1b right panel and Supplementary Table 2). Considering different reaction conditions used in previous studies, the performance of methanolysis has been divided into four categories (Figure 1b): (1) mixed solvent (tetrahydrofuran/chloroform)-assisted methanolysis; (2) homogeneous catalytic methanolysis; (3) heterogeneous methanolysis; and (4) heterogeneous layer-assisted methanolysis. The heterogeneous layer-assisted process illustrates the space-time yield ($g_{PET} g_{cat}^{-1} h^{-1}$)."

Revised Figure 1:

Figure 1. Catalytic Depolymerization of PET Polyester. a, b Diagram of PET glycolysis and methanolysis over V_o -rich Fe/ZnO NSs and its catalytic performance in comparison to catalysts in the literature.

Comment 8: *If the authors intended to uncover “the interactions between solvent, catalyst, and polymer”, orthogonal experimental design can be considered in this work.*

Our Response: We thank the reviewer for the constructive suggestions. In our study, we conducted preliminary experiments for the selection of catalysts, solvents, and polyesters. Applying an identical experimental condition is crucial for the consistency of the PET

depolymerization (i.e., alcoholysis process) in our study investigated. Therefore, we did not utilize an orthogonal experiment design. Notably, our study mainly focuses on exploring interactions at the interface, including the changes of solvent adsorption on the catalyst surface, the role of the catalyst in the polymer depolymerization process, and the effect of solvent in promoting polymer depolymerization. To reveal these mechanisms, we utilized a combination of characterization and simulation approaches to reveal the underlying mechanisms, which have not been well elucidated in the literature.

Although we did not employ an orthogonal experimental design, we realized the need to further strengthen our analysis of the solvent, catalyst, and polymer interactions. Following the reviewer's suggestion, we have conducted the following experiments and revised corresponding sections of our manuscript to better elucidate the depolymerization mechanisms under different conditions, including:

(1) Solvent–catalyst interactions under an air or N₂ atmosphere.

We utilized the *in situ* attenuated total reflectance (ATR) Fourier-transform infrared spectra (FTIR) technique to study the changes in the functional groups of methanol at different reaction temperatures on the catalyst surface under an air atmosphere. Our analysis, illustrated in Figure R4, reveals the increase in the β interaction (1056 cm⁻¹) and γ interaction (1010 cm⁻¹), which are attributable to the formation of alkoxy bonds between the primary hydroxyl groups and metal oxides ^{7, 8}(*Nat. Commun.* 14, 4509 (2023); *Nat. Commun.* 13, 5467 (2022)). However, a decrease in $V_{as}(\text{OH})$ at 3600-3080 cm⁻¹ indicates the activation of primary hydroxyl groups in methanol. This result suggests that methanol adsorbed on the surface of our V_o -rich-Fe/ZnO catalyst forms a specific structure (Structure **1**, shown in Figure R5). In contrast, under an N₂ atmosphere (Figure R6), the intensity of the hydroxyl group (1031 cm⁻¹) on the catalyst surface remained unchanged, and the characteristic peaks of β interaction (1056 cm⁻¹) and γ interaction (1010 cm⁻¹) did not appear, indicating that the catalyst does not activate methanol in the absence of air.

Figure R4. *In situ* attenuated total reflectance (ATR) infrared spectra investigated the changes in methanol functional groups adsorbed on the V_o -rich Fe/ZnO NSs surface under air atmosphere.

Figure R5. Schematic diagram of structure **1**.

Figure R6. *In situ* attenuated total reflectance (ATR) infrared spectra investigated the changes in methanol functional groups adsorbed on the V_o -rich Fe/ZnO NSs surface under the N_2 atmosphere.

The change of methanol activation energy barrier adsorbed on the V_o -rich Fe/ZnO NSs surface was further analyzed through theoretical calculation, as depicted in Figure R7. In V_o - Zn^{2+} - O - Fe^{3+} , the localized oxygen vacancy structure (denoted as **R**), serves as an anchoring site for an oxygen molecule, promoting the formation of the **M₁** species. This interaction thus activates the hydroxyl hydrogen of the methanol (CH_3OH) molecule adsorbed on **M₁**, facilitated by the oxygen molecule, resulting in the formation of **M₂** species. Then, the CH_3OH of **M₂** interacts with the V_o - Zn^{2+} - O - Fe^{3+} to form a transition state (denoted as **TS**) characterized by a metal alkoxy bond, which eventually leads to the formation of nucleophilic species in the **M₃** structure.

Figure R7. Free energy profiles for the conversion of $\text{O}_2 \rightarrow \text{O}_2^*$ and $\text{CH}_3\text{OH} + \text{O}_2^* \rightarrow \text{OOH}^* + \text{CH}_3\text{OH}^*$ on bulk ZnO and V_o -rich Fe/ZnO NSs.

(2) Depolymerization mechanisms under an air or N_2 atmosphere.

Then we synthesized **Modes 1 and 2** to simulate dimers and used *in situ* FTIR to study the changes in functional groups during the depolymerization of these **Modes** into monomers within the **M₃** structure. The specific analysis utilized *in situ* high-temperature-pressure infrared spectrometric (*in situ* HTP-IR) techniques were carried out at a synchrotron source, as detailed in Figure R8. This approach enabled us to delineate the depolymerization pathways by detecting intermediates and assessing isotope effects during the reactions.

As shown in Figure 5b, depolymerization in **Mode 1** with CH_3OH efficiently produces various intermediate species, including:

- 1) $\nu(\text{C}=\text{O})$ at 1730 cm^{-1} ;
- 2) $\nu(\text{C}-\text{O})$ at 1440 and 1259 cm^{-1} ;
- 3) $\nu(\text{O}-\text{C}-\text{H})$ at 1338 cm^{-1} ;
- 4) $\nu(\text{C}-\text{OH})$ at 1130 cm^{-1} ;
- 5) $\nu(\text{C}-\text{H})$ at 725 cm^{-1} .

The conversion of a hydroxyl group (1083 cm^{-1}) into triply bridged hydroxyl groups (1022 , 1080 , and 1118 cm^{-1}) indicates the formation of alkoxy bonds between hydroxyl groups and the catalyst (Figure 5b). The increasing intensity of $\nu(\text{C}-\text{O})$ at 1440 cm^{-1} and $\nu(\text{O}-\text{C}-\text{H})$ at 1338 cm^{-1} suggests the activation of the hydroxyl group in CH_3OH . Moreover, the peak intensities of these intermediates (C=O, C-O, and C-H) gradually rise, implying the transformation of **Mode 1** into DMT during the methanolysis process.

In comparison, *in situ* FTIR spectra showed that the OD absorbance band at 2500 cm^{-1} progressively increased and eventually stabilized upon adding CD_3OD (Figure 5c). As the reaction progresses, the OH bonds ($3400\text{--}3700 \text{ cm}^{-1}$) on the surface are

progressively increased, leading to the generation of ethylene glycol (EG) *via* **Mode 1** depolymerization. Taken together, these findings underscore the role of vacancies in promoting oxygen dissociation and transfer, which are critical processes in the formation of the **M₃** structure formation. Consequently, the interaction of Zn and CH₃OH* within the **M₃** structure activates the carbonyl groups (O=C–O) of **Mode 1** and promotes C–O bond cleavage, resulting in the formation of DMT and ethylene glycol.

Figure R8. *In situ* high-temperature-pressure infrared (*in situ* HTP-IR) spectra were investigated by **Mode 1** depolymerization in the V_o -rich Fe/ZnO NSs surface under **a)** air and **b)** N₂ atmosphere.

DFT calculations were employed to further analyze the changes in the energy barrier changes during the transformation of the **M₃** structure into monomers in **Mode 1** depolymerization, as depicted in Figure R9. The process begins with the initial **M₃** structure, designated as **R₁**. Here, Zn and CH₃OH* on the **M₃** structure activate the carbonyl group (O=C–O) of **Mode 1**. This activation leads to the formation of an intermediate species, **TS₁**, which subsequently adsorbs onto the **M₃** structure. Following adsorption, the C–O bond of **Mode 1** is cleaved, resulting in the generation of the **M₄** product and monomeric DMT. The subsequent step involves the initial **M₃** structure, **R₂**, where Zn and CH₃OH* on the **M₃** structure activate the carbon group (O=C–O) of the 1-(2-hydroxyethyl) 4-methyl terephthalate (HEMT) product. This interaction forms another intermediate species, **TS₂**, which then adsorbs onto the **M₃** structure. The cleavage of the C–O bond in the **M₅** product follows, leading to the production of ethylene glycol and DMT.

Figure R9. d-f Free energy profiles for **Mode 1** on bulk ZnO, and V_o -rich Fe/ZnO NSs, where 5f is the atomic structure change of **Mode 1** on the V_o -rich Fe/ZnO NSs.

Following the reviewer's comments, we have adjusted the content in the figures (Figure 3 and Figure 5). We have also revised the corresponding sections in the manuscript for clarification. The revised text and figures include:

Lines 247-263: “Besides the role of the oxygen vacancy sites, we conducted in situ experiments using attenuated total reflectance (ATR) Fourier transform infrared spectroscopy (FTIR) to elucidate reaction mechanisms of the solvent (i.e., methanol, CH_3OH) in *dehydrogenation over the $Zn^{2+}-V_o-Fe^{\delta+}$ pair activation of V_o -rich Fe/ZnO NSs surface* (Figure 3c). The in situ measurements were operated by gradually increasing the temperature from 25 to 160 °C and collecting spectra at ~ 10 °C intervals. *The OH stretching at $3,520\text{ cm}^{-1}$ provides information for assessing the possible contribution of hydroxyl intermediates.* During the reaction (Figure 3c), we observed *terminal hydroxyl groups (type I OH; 3697 cm^{-1}) and bridged hydroxyl groups (type II OH; 3360 cm^{-1}) on V_o -rich Fe/ZnO NSs (Figure 3c)⁴⁰. The terminal the $V_{as}(OH)$ signal at $3600\text{--}3080\text{ cm}^{-1}$, corresponding to the hydroxyl group was attributed to the defect driven $Fe-V_o-Zn-OH$ structure, gradually decreased, suggesting an association the activation of the higher type I OH with substantial deprotonated methanol (CH_3O^*). Type II OH, on the other hand, was assigned to the a primary hydroxyl group in methanol⁴³. Moreover, with increasing temperature (Figure 3c), we observed the transformation of the hydroxyl group (at 1030 cm^{-1}) into triply bridged hydroxyl groups (at 1004 , 1030 , and 1056 cm^{-1}). Such transformation indicated the formation of alkoxy bonds between hydroxyl groups and Fe-Zn, thus providing evidence of the activation of hydroxyl groups in CH_3OH Such results indicate that the β interaction (1056 cm^{-1}) and γ interaction (1004 cm^{-1}), attributed to the alkoxy bond between primary hydroxyl and metal*

oxides, gradually increase^{44, 45}. Compared with the N_2 atmosphere (Supplementary Figure 14), the intensity of the hydroxyl group (1030 cm^{-1}) adsorbed on the catalyst surface remained unchanged, and the characteristic peaks of the β interaction (1056 cm^{-1}) and γ interaction (1004 cm^{-1}) did not appear, indicating that methanol was not activated by catalyst under nitrogen atmosphere.”

Added Revised Figure 3c to the Manuscript:

Figure 3c. In situ attenuated total reflectance (ATR) infrared spectra investigated the changes in methanol functional groups adsorbed on the V_o -rich Fe/ZnO NSs surface under air atmosphere.

Added a new figure (Figure S14) to the Supporting Information:

Supplementary Fig. 14. In situ attenuated total reflectance (ATR) infrared spectra investigated the changes in methanol functional groups adsorbed on the V_o -rich Fe/ZnO NSs surface under N_2 atmosphere.

Lines 282-298: “In addition, to assess the ~~CH_3OH dehydrogenation over the $Zn^{2+}-V_o-Fe^{\delta+}$ pair~~ change of methanol activation energy barrier adsorbed on the V_o -rich Fe/ZnO NSs surface, we

constructed an energy diagram of the reaction pathways (Figure 3d), including the formation of $O_2 \rightarrow O_2^*$, as well as the determination of the activation energies of $CH_3OH + O_2^* \rightarrow OH^* + CH_3O^*$ (target product) $- O_2^* \rightarrow OOH^* + CH_3OH^*$. V_o - Zn^{2+} - O - Fe^{3+} localized oxygen vacancy structure (**R**) anchors an oxygen molecule to form **M₁** species. The hydroxyl hydrogen of the methanol (CH_3OH) molecule adsorbed on **M₁** is activated by oxygen molecules to form **M₂** species. Then CH_3OH of **M₂** and V_o - Zn^{2+} - O - Fe^{3+} form a transition state (**TS**) with a metal alkoxy bond, finally leading to the formation of the **M₃** structure of the nucleophilic species. Notably, the activation energies of the transition state (**TS**) activation revealed that bulk ZnO (0.55 eV) (Supplementary Figure 16 and Supplementary Table 6) ~~exhibits elevated activation energies for continuous methanol dehydrogenation, whereas~~. In contrast, V_o -rich Fe/ZnO NSs exhibit low activation energies (0.21 eV) (Figure 3d, Supplementary Figure 17 and Supplementary Table 7). This result indicates that the V_o -rich Fe/ZnO NSs possess a higher activity for ~~the dissociation of $O-O_2 \rightarrow O_2^*$ and $H-O$ bond to form O^* and CH_3O^*~~ $CH_3OH + O_2^* \rightarrow OOH^* + CH_3OH^*$ species, essential for the subsequent C–O disconnection of PET depolymerization. These findings align with the observations using in situ FTIR, underscoring that V_o -rich Fe/ZnO NSs featuring V_o - Zn^{2+} - O - Fe^{3+} are highly effective in catalyzing the activation of CH_3OH . ”

Added revised Figure 3d to the Manuscript:

Figure 3d. Free energy profiles for the conversion of $O_2 \rightarrow O_2^*$ and $CH_3OH + O_2^* \rightarrow OOH^* + CH_3OH^*$ on bulk ZnO and V_o -rich Fe/ZnO NSs.

Lines 360-368: “...As the reaction progresses, the OH bonds ($3400\text{--}3700\text{ cm}^{-1}$) on the surface are progressively increased to the generation of EG by **Mode 1** depolymerization (Supplementary Figure 22a, b). ~~These results suggest that the V_o -rich Fe/ZnO NSs efficiently convert CH_3OH into the CH_3O^* intermediate via dehydrogenation, thus accelerating the depolymerization of PET. It is worth noting that vacancies on catalysts could facilitate hydrogen dissociation and transfer, as demonstrated in the literature.~~ In contrast, under the nitrogen atmosphere, the in situ IR spectrum of the depolymerization reaction does not exhibit the characteristic functional group changes mentioned above (Supplementary Figure 23). Taken together, these vacancies can promote oxygen dissociation and transfer, which are critical for

the formation of the M_3 structure. Thus, Zn and CH_3OH^* on the M_3 structure activate the carbon groups ($O=C-O$) of **Mode 1** and promote C–O bond breaking to form DMT and ethylene glycol.”

Added a new Figure S23 to the Supporting Information:

Supplementary Fig. 23. In situ high-temperature-pressure infrared spectrometry was investigated by **Mode 1** depolymerization under nitrogen.

Lines 370-397: “To further understand the effect of oxygen vacancy in promoting the ~~C–O cleavage~~ depolymerization of PET, we conducted DFT calculations for the depolymerization of PET on V_o -rich Fe/ZnO NSs and bulk ZnO. ~~Calculations results revealed that the $Zn^{2+}-V_o-Fe^{\delta+}$ pair, in conjunction with the CH_3O^* intermediate, activates the ester bond of PET with the highest free energy (TS) in the rate determining step (RDS). The energy barrier (TS1) for C–O bond cleavage on bulk ZnO [$\Delta G = 0.08$ eV]. (Figure 5c, Supplementary Figure 13 and Supplementary Table 8) and V_o -rich Fe/ZnO NSs [$\Delta G = 0.12$ eV] (Figure 5c, Supplementary Figure 14 and Supplementary Table 9). The initial structure R_1 represents the M_3 structure. Zn and CH_3OH^* on the M_3 structure activate the carbon group ($O=C-O$) of **Mode 1** to form an intermediate species (TS_1). On the bulk ZnO surface, the formation of TS_1 necessitates overcoming an activation energy of 0.08 eV (Figure 5d, and Supplementary Figure 24). Detailed data is provided in the SI (Supplementary Table 8). In contrast, on the V_o -rich Fe/ZnO NSs surface, the calculated barrier for the formation of TS_1 decreases to -0.12 eV, substantially lower than that on bulk ZnO (Figure 5d, f, Supplementary Figure 25). Detailed data is also provided in the SI (Supplementary Table 9). These results underscore that the presence of the oxygen vacancy facilitates methanol and **Mode 1** adsorption on the catalyst surface. This intermediate species is then adsorbed on the M_3 structure, where the C–O bond of **Mode 1** is broken, generating species M_4 product and monomeric DMT.~~”

~~These results support that the oxygen vacancy rich Fe/ZnO NSs are the active sites for the selective C–O disconnection to 1-(2-hydroxyethyl) 4-methyl terephthalate (HEMT). In the latter~~

stages, as depolymerization proceeded, the process shifted from the HEMT formation to DMT generation, indicating that limited DMT yield and selectivity was the C–O bond breaking of the HEMT. Similarly, the initial structure R_2 represents the M_3 structure. Zn and CH_3OH^ on the M_3 structure activate the carbon group ($O=C-O$) of the 1-(2-hydroxyethyl) 4-methyl terephthalate (HEMT) product to form an intermediate species TS_2 . On the bulk ZnO surface, the C–O bond cleavage of the HEMT necessitates surmounting the activation energy barrier (TS_2) of 0.27 eV (Figure 5e). On the V_o -rich Fe/ZnO NSs surface, the TS_2 of -0.35 eV is lower than bulk ZnO (Figure 5e, f). The energy barrier (TS_2) of C–O bond cleavage of the HEMT on V_o -rich Fe/ZnO NSs [$\Delta G = -0.35$ eV] lower than bulk ZnO [$\Delta G = 0.27$ eV] (Figure 5d), suggesting the formation of CH_3O^* species and O^* species in the $Zn^{2+}-V_o-Fe^{\delta+}$ pair significantly enhances the cleavage ability of C–O, accelerated for ester bond activation and C–O bond cleavage. This enhancement thus leads to the low activation energies for ester bond activation and C–O bond cleavage on V_o -rich Fe/ZnO NSs. This result underscores that the formation of CH_3OH^* species and OOH^* species in the $V_o-Zn^{2+}-O-Fe^{3+}$ sites significantly enhances the cleavage ability of C–O, thereby accelerating ester bond activation and C–O bond cleavage. This species is then adsorbed on the M_3 structure, breaking the C–O bond of the M_4 product and thus generating ethylene glycol and DMT. Taken together, V_o -rich Fe/ZnO NSs promote the formation of the nucleophilic species M_3 structure from the adsorbed methanol and activate the carbon group of PET. This synergistic action catalyzes the breaking of the C–O in the ester bond, leading to the generation of DMT and ethylene glycol.”*

Added revised Figure 5d to the Manuscript:

Figure 5d-f. Free energy profiles for **Mode 1** on bulk ZnO, and V_o -rich Fe/ZnO NSs, where 5f is the atomic structure change of **Mode 1** on the V_o -rich Fe/ZnO NSs.

Comment 9: *The LCA procedure lacks basic elements (e.g., four basic steps) and sufficient life cycle inventory. Therefore, the LCA analysis seems incomplete and unconvincing.*

Our Response: Thank the reviewer for the valuable suggestions regarding our life cycle assessment (LCA) analysis. We understand the importance of including all foundational elements and providing a thorough inventory to support our LCA claims.

In response to the reviewer’s comment, we wish to clarify that we indeed provide details of our LCA procedure in the Supplementary Information. These details include the four basic steps—goal and scope definition, inventory analysis, impact assessment, and interpretation. This information can be found in Supplementary Figures 28-29 and Figure R10. Additionally, a detailed cycle inventory is presented in Supplementary Table 11, which evaluates the environmental impact of PET recycling.

The system boundary of LCA is defined as “cradle to gate”, including collection and transportation of waste PET, production of catalysts, and stages of post-depolymerization and re-polymerization. We assume that the pretreatment process for PET waste plastic aligns with the standard protocols of mechanical recycling. In terms of allocation, our study adopts the “cut-off” rule, which distinctly separates the life cycle of the original plastic from that of the recycled plastic, allowing for an independent evaluation of each. Notably, the material and energy balance was derived from actual experimental data, supplemented by simulations conducted in Aspen Plus. These elements form the basis of our proposed prospective processes, which are modeled to reflect prospective industrial scenarios. The functional unit for this LCA is defined as 1 kg of amorphous PET resin (PRO-PET in Aspen Plus).

Figure R10. Schematic illustration of closed-loop recycling PET *via* methanolysis.

In response to the reviewer’s comments and to ensure that both readers and reviewers to have a clear understanding of the LCA results, we have added the corresponding content to the manuscript.

Lines 434-442: “*The PET waste recycling technology boundary mainly included: (1) mechanical shredding of waste PET (collection, transportation, and pretreatment); (2) PET depolymerization (catalyst synthesis, PET methanolysis/separation); (3) re-polymerization; and (4) r-PET of extrusion. Detailed information on the LCA approach is provided in the SI (Supplementary Table 13, and Supplementary Figures 28-29). The chemical recycling processes were simulated on an industrial scale with an annual treatment of 200,000 tons of waste PET, using Aspen Plus V11 to obtain the mass balance and energy consumption.*”

Comment 10: *Detailed processing parameters can be used to replace vague words. For instance, in Page 16, the authors need to specify the number of rpm, instead of using “vigorous stirring”. Further, I recommend that references can be added to support the selection of processing parameters.*

Our Response: We appreciate the reviewer’s emphasis on precision and reproducibility in experimental descriptions. To address the reviewer’s comment, we have updated the manuscript to replace vague descriptions with precise processing parameters. Although the stirring speeds in zinc oxide synthesis are not explicitly detailed in the foundational literature⁹⁻¹¹(*J. Am. Chem. Soc.* 2009, 131, 12540–12541; *J. Am. Chem. Soc.* 2016, 138, 2225–2234), we have determined from our experiments that a stirring speed range of 800–1000 rpm ensures consistent synthesis outcomes. This range has been validated to maintain the repeatability and effectiveness of the material synthesis across various batches during our experiment. Accordingly, the term “vigorous stirring” on Page 16 has been replaced with “stirring at 800–1000 rpm”. We believe that specifying these details will enhance the clarity and reproducibility of the experimental procedures outlined in our study. The revised text now reads:

Lines 497-506: “*Zinc chloride anhydrous (40 mmol), ferric chloride (0.8 mmol), and L-alanine (4 mmol) were dissolved in 125 mL of ethanol and deionized (DI) water (v/v=1/4) and vigorously stirred (800 rpm) until a homogeneous solution was formed. Ethanolamine (5 mL) was dissolved in 60 mL of DI water/ethanol (v/v=1/2). Then, the mixed solution was added dropwise into the mixed solution under vigorous stirring (800 rpm) at 25 °C for 6 h.*”

Lines 508-510: “*Sodium hydroxide solution (0.08 mol/L, 50 mL) was slowly dropped into the mixed solution under vigorous stirring (800 rpm) at 180 °C for 6 h.*”

REVIEWER #3

Before our response to the specific comments of the reviewer, we would like to highlight the key revisions made:

1. Revising the Introduction section to highlight the challenges of existing PET recycling strategies and emphasize the novelty and scientific contribution of our study.
2. Adding detailed descriptions of our experimental procedures, including depolymerization processes, catalyst regeneration procedures, and chromatography testing conditions.
3. Conducting *in situ* FTIR measurements of PET depolymerization under a nitrogen atmosphere, providing insights into the role of oxygen species in the catalyst-mediated activation of methanol and depolymerization.
4. Performing *in situ* EPR measurements of PET depolymerization at various reaction temperatures and atmospheres (Figure 3b).
5. Adding a new figure (Figure S30) that includes all 18 indicators evaluated in our comprehensive life cycle assessment of PET depolymerization processes.

General Comment: *The authors synthesize a PET alcoholysis catalyst consisting of nanosheets of ZnO and Fe. They tested a wide range of waste PET and plastic mixtures and showed the catalyst gave very high conversions and yields when impurities were present. The DMT monomer could successfully be repolymerized to PET showing the possibility of closed-loop recycling. They characterized the catalyst, then performed DFT-informed *in situ* IR experiments, and deuterium-labeling studies. An LCA was included showing the process could improve the sustainability relative to virgin PET production.*

This is a topic of much interest recently and the range of materials tested, and the effectiveness of the catalyst are both impressive. It is also appropriate to include many of the characterization techniques used and the inclusion of an LCA is also appropriate and useful. However, the purpose of the experiments are in general not well explained, there are numerous inconsistencies, and many important details are not included. Therefore, I recommend this manuscript should be rejected, although there are many suggestions included below the authors may consider to improve this piece of research.

Our Response: We sincerely appreciate the time and efforts the reviewer has invested in evaluating our manuscript and providing detailed feedback. We recognize the concerns about the clarity of experiment purposes, inconsistencies, and missing details in the manuscript. To address these issues, we have undertaken a thorough revision of the manuscript to enhance the clarity and completeness of our study.

In this revision, we have clarified the objectives of each experiment and resolved inconsistencies throughout the text. We have also incorporated additional experimental

results and more detailed characterization to fully demonstrate the effectiveness and sustainability of our synthesized catalyst in polyester waste recycling. In our LCA analysis, we have now expanded our analysis to cover all potential impacts of the recycling process, assessing effects on human health, ecosystems, and resource use across 18 evaluated categories. These comprehensive LCA evaluations provide a robust framework for understanding the environmental advantages of our recycling approach.

We believe these revisions and additions address the reviewer's concerns and significantly strengthen the manuscript, making a compelling case for the contribution of our research to sustainable PET plastic recycling technologies. Below, we provide a detailed point-by-point response to the reviewer's comments.

Comment 1: *Line 35-36. Specify that PET can be recycled into BHET or DMT. The end product depends on the reaction.*

Our Response: We agree with the reviewer's statement that specific products of PET chemical recycling are indeed dependent on the nature of the reaction process. To clarify this point in the Introduction, we have revised the text in Lines 35-36 to more accurately reflect that PET can be depolymerized through alcoholysis processes (i.e., methanolysis and glycolysis) to yield products such as dimethyl terephthalate (DMT) and bis(2-hydroxyethyl) terephthalate (BHET). The modified statement now explicitly states that the outcome of PET recycling varies based on the chosen alcoholysis method, leading to the production of either DMT or BHET. To address the reviewer's comments, we further revised the manuscript for clarification. The revised text now reads:

Lines 34-37: *~~“Chemical recycling of poly(ethylene terephthalate) (PET) waste to produce high-value monomers, such as bis-2-hydroxyethyl terephthalate (BHET) and dimethyl terephthalate (DMT);~~ Alcoholysis of poly(ethylene terephthalate) (PET) waste to produce monomers, including methanolysis to produce dimethyl terephthalate (DMT) and glycolysis to produce bis-2-hydroxyethyl terephthalate (BHET), is a promising strategy in PET waste management.”*

Comment 2: *Lines 65-68. Most alcoholysis operates with a metal salt catalyst, not with strong acids or bases or under supercritical conditions. The description provided could be misleading and should be changed to reflect the general practice of the field.*

Our Response: We appreciate the reviewer's attention to detail and agree that our initial description of the alcoholysis process could be refined to reflect general practices in the field more accurately.

Most of the alcoholysis process uses homogeneous metal salts, ionic liquids, enzymes, and other catalysts. Although the catalyst achieves adequate efficiency, challenges such as

difficulty in separating the product, leaching of metal ions, or catalyst instability remain prevalent.

To address the reviewer's comments, we further revised the manuscript for clarification. The revised text now reads:

Lines 63-70: ~~“However, this alcoholysis largely operates under harsh conditions, including the use of caustic strong bases and acids, which cause severe equipment corrosion and generate hazardous liquid waste²⁸, or under high temperatures (e.g., >200 °C)^{29,30}. Such challenges limit the real applications of these chemical recycling methods in PET waste management. Hence, there is a crucial need to develop a green, efficient, and sustainable alcoholysis method for converting real PET wastes into monomer units. Alcoholysis is a process that utilizes alcohol as a solvent to depolymerize PET into monomers. The efficacy of this process largely relies on the design of the catalyst. Previous studies on this process typically involved the use of catalysts like homogeneous metal salts, ionic liquids, enzymes, and other catalysts²⁸⁻³⁰. While these catalysts achieve adequate efficiency, challenges such as difficulty in separating the product from the catalyst, leaching of metal ions, or catalyst instability remain prevalent. Consequently, developing a green, efficient, and sustainable catalyst for alcoholysis to convert PET into monomers is imperative.”~~

Comment 3: Lines 77-83. The listed needs do not directly lead to the conclusion of needing new catalysts. The list of challenges should be amended to include higher activity catalysts.

Our Response: We sincerely thank the reviewer for the insightful comment. We acknowledge the importance of clearly connecting the challenges listed to the necessity for developing new catalysts, particularly those with higher activity. We have re-examined Lines 77-83 of the manuscript and made significant revisions to better articulate this connection.

In the revised manuscript, we now explicitly address the limitations of current catalysts, initially discussed in Lines 71-94, and highlight the ongoing research into two types of highly active catalyst types in the alcoholysis process. We have also amended the section in Lines 71-94 to more clearly outline the critical challenges in the catalyst and alcoholysis process, emphasizing the urgent need for catalysts that not only overcome existing problems but also offer enhanced activity and efficiency. The revised text now reads:

Lines 71-94: ~~“While these studies have reported some promising results from lab-scale experiments, scaling up the developed methods into real applications is hindered by the following challenges³⁴⁻³⁶. First, PET methanolysis pathways and mechanisms are poorly understood, particularly regarding the interactions between solvent, catalyst, and polymer. Second, efforts on recycling real PET plastics, as well as validating the effectiveness of monomer reuse, are still limited. Third, a comprehensive life cycle assessment for the entire process of PET alcoholysis is also lacking. Therefore, innovative design of catalysts, as well as insights into reaction mechanisms, are critically needed for closed-loop recycling of real PET waste~~

~~depolymerization.~~ A recent study introduced the concept of creating a solid-solid interface between plastic and catalyst, allowing methanolysis of the polymer through methanol vapor²⁶. This innovative approach to interfacial catalysis opens up new avenues for the depolymerization of plastic waste. Additionally, other studies also demonstrate the promise of using oxygen vacancy-metal oxide in catalyzing the depolymerization of PET to produce BHET monomers³¹⁻³³. For instance, defect-rich CeO₂ nanoparticles modified with KH550 have been used for the glycolysis of PET. , at the critical boiling point of ethylene glycol (196 °C), a PET conversion rate of 98.6% with a BHET yield of 90.3% was achieved using this approach³⁴.

Despite these encouraging findings, there are limited efforts on validating catalysts for real polyester waste, such as cotton blends, textiles, and undegraded carpets, leaving several challenges unaddressed³⁵⁻³⁷. First, most articles focus solely on catalyst design and PET depolymerization efficiency, with few investigating the pathways and mechanisms of PET depolymerization. This knowledge gap exists because the alcoholysis process involves high-temperature, solid-liquid-solid complex interface reactions among catalysts, polymers, and solvents. Moreover, there is a lack of research on the interactions between catalysts and solvents, solvents and polymers, and catalysts and polymers during the reaction process. Second, few studies examine the recycling of actual PET plastics or textiles, which contain additives like pigments and plasticizers that can affect the catalytic efficiency. Third, while PET recycling aims to recover waste, comprehensive life cycle evaluations, as well as economic and technical analysis of the entire recycling process, are scarce. Therefore, innovative catalyst design and deeper insights into reaction mechanisms, are critically needed for closed-loop recycling of real PET waste.”

Comment 4: Line 84. “NS” needs to be defined as nanosheets before using the abbreviation.

Our Response: We appreciate the reviewer’s attention to detail. To address the reviewer’s comment, we have now ensured that "NS" is properly defined as "nanosheets" at the first point of use within the manuscript. The revised text now reads:

Line 95: “Here, we synthesize an oxygen vacancy (V_o)-rich Fe/ZnO nanosheets (NSs) catalyst for polyester plastic depolymerization.”

Comment 5: Lines 130-131. Grammar

Our Response: We thank the reviewer for the careful review. We have addressed the comment in the Lines 148-149 in manuscript. The revised text now reads:

Lines 148-149: “Further increasing the temperature to 160 °C enabled efficient depolymerization of PET to generate DMT. ~~effectively depolymerized results efficient depolymerization~~”

Comment 6: Lines 133-134. Why not run the PET/PC reaction at 160C and depolymerize everything simultaneously? Mechanical sorting is cited as the issue being addressed here but surely the stepwise reaction is meant to allow for distillation of BPA from unreacted PET to avoid chemical separations of the mixture of products, not mechanical separations. Please clarify.

Our Response: Thank the reviewer for the valuable suggestion. We appreciate the opportunity to clarify this point. The challenge with mechanically sorting PET/PC mixed plastics arises from their similar densities (PC: 1.2 g/cm³, PET: 1.3 g/cm³). To address this challenge, we designed a selective chemical depolymerization process. Initially, we conducted a methanolysis experiment on PET/PC mixed plastics at 120 °C. The results demonstrated that PC was completely converted with a bisphenol A (BPA) yield greater than 99%, while PET remained unreacted. This phenomenon allows for the separation of BPA from unreacted PET. Subsequently, the unreacted PET was subjected to methanolysis at 160 °C, resulting in complete conversion to dimethyl terephthalate (DMT) with a yield greater than 99%.

To explore the feasibility of simultaneous depolymerization, we also performed a methanolysis experiment at 160 °C on the PET/PC mixture. The results indicated that both PET and PC were completely converted, producing a mixture of DMT and BPA. However, separating this mixture proved challenging (Figure R1).

In summary, our stepwise selective chemical depolymerization approach effectively addresses the difficulties of mechanical sorting and separation of mixed depolymerized products. By sequentially converting PC and PET at different temperatures, we achieve efficient recycling of PET/PC mixed plastics into their monomers, DMT and BPA, thereby simplifying the separation process.

Figure R1. Schematic diagram of PET/PC mixed plastics through selective chemical depolymerization.

Following the reviewer's comments, we further revised the Lines 152-155 in manuscript for clarification. The revised text now reads:

Lines 152-155: "... ~~Given the challenging nature of the mechanical separation of mixed polyester plastic wastes, this could present a useful strategy for recovering valuable products.~~

This approach enables the selective chemical depolymerization of PET/PC mixed plastics to yield monomers, i.e., DMT (from PET) and BPA (from PC), effectively overcoming the challenges associated with mechanical sorting and separating mixed depolymerization products.”

Added a new figure (Figure S6) to the Supporting Information:

Supplementary Fig. 6. *Schematic diagram of PET/PC mixed plastics through selective chemical depolymerization.*

Comment 7: *Line 147. The authors should clarify what is meant when the catalyst possesses structural variables.*

Our Response: We appreciate the reviewer's careful suggestion. We have completed the revision of Lines 166-168 in the manuscript to clarify the claim about the catalyst structural variables. The revised text now reads:

Lines 166-168: *“Moreover, the V_o -rich Fe/ZnO NSs catalyst used in polyester depolymerization exhibits structural stability ~~variables and high reusability~~. It remained high activity and selectivity after 5 cycles, demonstrating consistent performance.”*

Comment 8: *Line 150. The catalyst regeneration process used for catalyst recyclability tests should be noted somewhere. I do not see it anywhere.*

Our Response: We thank the reviewer for the comments. The specific steps for catalyst regeneration are as follows. The reacted catalyst was stirred in hot ethanol (70 °C) for 20 minutes at 300 rpm, then filtered to collect the filter cake layer and dried at 60 °C for 8 hours. The resulting product was subsequently transferred to a tube furnace for calcination at 350 °C for 1 hour in an atmosphere of 1 vol H₂ /99 vol N₂, with a heating rate 2 °C /min.

In response to the reviewer's comments, we have added the above regeneration process to the supplementary methods section of the SI for clarification. The revised text in the SI now reads:

Supporting Information, Page S2, Lines 56–59: *“The reacted catalyst was stirred in hot ethanol (70 °C) for 20 minutes at 300 rpm, then filtered to collect the filter cake layer, dried at*

60°C for 8 h. The resulting product was subsequently transferred to a tube furnace for calcination at 350°C for 1 h in an atmosphere of 1 vol H₂ /99 vol N₂, with a heating rate of 2°C /min.”

Comment 9: In Figure 2a, the two depicted oxygen vacancies are not drawn the same. There is a 2-coordinate Zn near the left vacancy while the same location has a 3-coordinate Zn near the right vacancy.

Our Response: We agree with the reviewer's observation regarding the inconsistencies in the depiction of oxygen vacancies in Figure 2a. Indeed, in the old schematic diagram, a line representing Zn–O bond was inadvertently omitted. To provide a clearer representation of our structure, we have replaced the original schematic with the structure derived from our calculation process, as shown in the revised Figure 2a.

Revised the schematic in Figure 2a in the manuscript:

Figure 2a. Schematic illustration of the synthesis of V_o-rich Fe/ZnO NSs.

Comment 10: Lines 192-193. The authors make a comparison the “other electronic slabs.” This is in reference to the difference between a surface layer vs the second layer deep. It is suggested to simplify the language and say, “the second layer” instead of “other electronic slabs.”

Our Response: We deeply appreciate the reviewer’s effort in improving the quality of our study. We agree that the term “second layer” is clearer and more appropriate than “other electronic slabs”. We have made this change in the manuscript at Lines 192-193 for better clarity. The revised text now reads:

Lines 227-229: “Notably, the oxygen vacancy sites on a top slab of ZnO (100) surfaces were more readily generated ($\Delta E = 3.55$ eV) (~~$\Delta E = 3.53$ eV~~) (Figure 3a and Supplementary Table 4), when compared to ~~other electronic slabs~~ the second layer slab.”

Comment 11: Lines 193-194. The presence of five slabs of Fe/ZnO gave enhanced vacancy generation. Please state what this was compared to.

Our Response: Thank the reviewer for the valuable suggestions. To explore the most stable structure of Fe atom-doped ZnO (100) with oxygen vacancies, we calculated the formation energy of oxygen vacancies in various positions (D1 to D11 as shown in Supplementary Fig. 13). The results demonstrate that the fifth configuration (D5) possesses the lowest defect formation of 3.535 eV, which is even lower than that in the pure ZnO slab.

With the reviewer's comments, we revised Lines 193-194 to add the necessary clarification. The revised text now reads:

Lines 229-234: ~~“The presence of five slabs of Fe atom-doped ZnO (100) surfaces significantly enhances oxygen vacancy generation ($\Delta E = -3.55$ eV) (Figure 3a and Supplementary Table 5). These findings demonstrate that a defective Fe/ZnO (100) surface is more likely to be obtained and it is thus expected to exhibit better performance in the formation of oxygen-active species. To explore the most stable structure of Fe atom-doped ZnO (100) with oxygen vacancies, we calculated the formation energy of oxygen vacancies at various positions (D1 to D11, shown in Supplementary Fig. 13). The results demonstrate that the fifth configuration (D5) has the lowest defect formation of 3.535 eV, which is even lower than that in the pure ZnO slab.”~~

Comment 12: Line 201. The designation of iron in the $\text{Zn}^{2+}\text{-V}_\text{o}\text{-Fe}^{\delta+}$ with a delta should be explained.

Our Response: The reviewer raised an interesting point for our consideration. Following the reviewer's comment, we performed an additional experiment using X-ray photoelectron spectroscopy (XPS) to determine the valence state of $\text{V}_\text{o}\text{-Zn}^{2+}\text{-O-Fe}^{\delta+}$.

In the Zn 2p region (Figure R2a), Zn 2p_{1/2} and Zn 2p_{3/2} peaks appear at 1044.7 and 1021.6 eV, respectively, indicative of the +2 oxidation state of Zn. In the Fe 2p region (Figure R2b), Fe 2p_{1/2} and Fe 2p_{3/2} peaks appear at 724.9 and 711.3 eV, respectively, indicative of the +3 oxidation state of Fe. As such, $\text{V}_\text{o}\text{-Zn}^{2+}\text{-O-Fe}^{\delta+}$ was determined to be $\text{V}_\text{o}\text{-Zn}^{2+}\text{-O-Fe}^{3+}$. Additionally, in the O 1s region (Figure R2c), the peaks at 531.4 eV and 529.7 eV correspond to the O atoms in the vicinity of oxygen vacancies and the lattice oxygen of Zn–O–Fe, respectively.

Figure R2. High-resolution XPS spectra of the as-prepared V_o -rich Fe/ZnO NSs catalysts. XPS spectra of **a** Zn 2p, **b** Fe 2p, and **c** O 2p.

To address the reviewer's comments, we have clarified the designation of iron in the state of V_o -Zn²⁺-O-Fe³⁺ throughout the main manuscript (**line 247, line 272, line 287, line 296, line 390**). Additionally, we have added the above figures and analysis to the SI for clarification.

Added an XPS description to the characteristics of V_o -rich Fe/ZnO NSs catalyst manuscript: “We used X-ray photoelectron spectroscopy (XPS) characterization to determine the valence state of V_o -Zn²⁺-O-Fe^{δ+}. In the Zn 2p region (Supplementary Figure 11a), Zn 2p_{1/2} and Zn 2p_{3/2} peaks arise at 1044.7 and 1021.6 eV, respectively, indicative of the +2 oxidation state of Zn. In the Fe 2p region (Supplementary Figure 11b), Fe 2p_{1/2} and Fe 2p_{3/2} peaks arise at 724.9 and 711.3 eV, indicative of the +3 oxidation state of Fe. As such, V_o -Zn²⁺-O-Fe^{δ+} was determined to be V_o -Zn²⁺-O-Fe³⁺. Additionally, in the O 1s region (Supplementary Figure 11c), the peaks at 531.4 eV and 529.7 eV correspond to the O atoms in the vicinity of oxygen vacancies and the lattice oxygen of Zn-O-Fe, respectively⁴².”

Added a new figure (Figure S11) to the Supporting Information:

*Supplementary Fig. 11. High-resolution XPS spectra of the as-prepared V_o -rich Fe/ZnO NSs catalysts. XPS spectra of **a** Zn 2p, **b** Fe 2p, and **c** O 2p.*

Comment 13: Lines 203-204. *It is known that ESR sensitivity is inversely correlated to temperature, intrinsically. It cannot be said that the oxygen vacancy density changes when temperature changes without more careful standardization at the compared temperatures.*

Our Response: We agree with the reviewer that ESR sensitivity is inherently inversely correlated to temperature increase. To avoid errors caused by temperature and signal sensitivity in *in situ* testing, we standardized our approach by collecting catalysts after reactions at different temperature stages under both nitrogen and air atmospheres before testing with EPR, as shown in Figure R3.

Figure R3. *Quasi-in situ EPR spectra for the detection of the evolution of oxygen vacancies on V_o -rich Fe/ZnO NSs catalysts under air and nitrogen, respectively.*

In particular, we observe a symmetrical EPR peak at $g = 2.003$ in V_o -rich-Fe/ZnO, attributable to unpaired electrons associated with oxygen vacancies. Interestingly, the intensity of the peak progressively increases with rising reaction temperature under air, indicating an increase in oxygen vacancy density. Conversely, the peak intensity decreases with rising reaction temperature under N_2 , suggesting a decreased density of oxygen vacancy on V_o -rich Fe/ZnO NSs. Taken together, these observations indicate that O_2 activation is a significant factor in increasing the density of oxygen vacancy.

With the reviewer's comments, we have revised Lines 203-204 for clarification. The revised text now reads:

Lines 240-246: “Notably, the intensity of the peak ~~exhibits a gradual decline as reaction temperature increases, suggesting the decreased density of oxygen vacancy on V_o -rich Fe/ZnO NSs~~ shows a progressive increase with rising reaction temperature under air, indicating an increase in oxygen vacancy density. In contrast, the intensity of the peak progressively decreases

with rising reaction temperature under N₂, implying a decreased density of oxygen vacancy on V_o-rich Fe/ZnO NSs. Taken together, these results demonstrate the activation of O₂ (in air) is an important factor in increasing the density of oxygen vacancies.”

Revised the content in Figure 3b in the manuscript:

Figure 3b. *Quasi-in situ EPR spectra for the detection of the evolution of oxygen vacancies on V_o-rich Fe/ZnO NSs catalysts under air and nitrogen, respectively.*

Comment 14: *Lines 205-207. The author appears to be attributing the difference with oxygen vacancy concentration at higher temperatures to gradual adsorption of oxygen. How much of the actual signal loss is attributable to this effect vs a temperature-dependent oxygen vacancy stability vs some intrinsic ESR temperature-dependence? This data needs more clear explanation of its importance.*

Our Response: We would like to thank the reviewer for the valuable suggestion. We agree with the reviewer that attributing differences in oxygen vacancy concentration at high temperatures to the gradual adsorption of oxygen is insufficient. To avoid errors caused by temperature and signal sensitivity using *in situ* testing, we collected the catalysts after the reaction at different reaction temperature stages under nitrogen and air to test EPR, as shown in Figure R3. We observe a symmetrical EPR peak at $g = 2.003$ in V_o-rich-Fe/ZnO, attributable to unpaired electrons associated with oxygen vacancies. Notably, the peak intensity peak shows a progressive increase with rising reaction temperature under air, implying the increased density of oxygen vacancy on V_o-rich Fe/ZnO NSs. In contrast, the intensity of the peak shows a progressive decrease with rising reaction temperature under N₂, suggesting the decreased density of oxygen vacancy on V_o-rich Fe/ZnO NSs. These observations imply that O₂ activation significantly contributes to the increase in oxygen vacancy density. To further elucidate these findings, we recognize the need to differentiate the effects of oxygen adsorption from intrinsic ESR temperature dependence and oxygen vacancy stability at higher temperatures.

With the reviewer's comments, we revised the text in Lines 205-207 for clarification. The revised text now reads:

Lines 244-246: “~~Note that this measurement was conducted under ambient conditions. As a consequence, oxygen molecules in the air gradually adsorbed to the catalyst surface to occupy oxygen vacancy sites, thus leading to the observed phenomenon.~~ Taken together, these results demonstrate the activation of O₂ (in the air) is an important factor in increasing the density of oxygen vacancies.”

Comment 15: Lines 222-224. The claim that triply bridged hydroxyl groups indicates the formation of alkoxy bonds between hydroxyl groups and Fe-Zn is not obvious and deserves more explanation.

Our Response: Following the reviewer's comments, we utilized the *in situ* attenuated total reflectance (ATR) Fourier-transform infrared spectra (FTIR) technique to study the changes in the functional groups of methanol at different reaction temperatures on the catalyst surface under an air atmosphere. Our analysis, illustrated in Figure R4, reveals the increase in the β interaction (1056 cm⁻¹) and γ interaction (1010 cm⁻¹), which are attributable to the formation of alkoxy bonds between the primary hydroxyl groups and metal oxides (*Nat. Commun.* 14, 4509 (2023); *Nat. Commun.* 13, 5467 (2022); *Angew. Chem. Int. Ed.* 134, e202116059 (2022)). However, a decrease in $V_{as}(\text{OH})$ at 3600-3080 cm⁻¹ indicates the activation of primary hydroxyl groups in methanol. This result suggests that methanol adsorbed on the surface of our V_o -rich-Fe/ZnO catalyst forms a specific structure (Structure 1, shown in Figure R5). In contrast, under an N₂ atmosphere (Figure R6), the intensity of the hydroxyl group (1031 cm⁻¹) on the catalyst surface remained unchanged, and the characteristic peaks of β interaction (1056 cm⁻¹) and γ interaction (1010 cm⁻¹) did not appear, indicating that the catalyst does not activate methanol in the absence of air.

Figure R4. *In situ* attenuated total reflectance (ATR) infrared spectra investigated the changes in methanol functional groups adsorbed on the V_o -rich Fe/ZnO NSs surface under air atmosphere.

Figure R5. Schematic diagram of structure **1**.

Figure R6. *In situ* attenuated total reflectance (ATR) infrared spectra investigated the changes in methanol functional groups adsorbed on the V_o -rich Fe/ZnO NSs surface under the N_2 atmosphere.

The change of methanol activation energy barrier adsorbed on the V_o -rich Fe/ZnO NSs surface was further analyzed through theoretical calculation, as depicted in Figure R7. In V_o -Zn²⁺-O-Fe³⁺, the localized oxygen vacancy structure (denoted as **R**), serves as an anchoring site for an oxygen molecule, promoting the formation of the **M₁** species. This interaction thus activates the hydroxyl hydrogen of the methanol (CH₃OH) molecule adsorbed on **M₁**, facilitated by the oxygen molecule, resulting in the formation of **M₂** species. Then, the CH₃OH of **M₂** interacts with the V_o -Zn²⁺-O-Fe³⁺ to form a transition state (denoted as **TS**) characterized by a metal alkoxy bond, which eventually leads to the formation of nucleophilic species in the **M₃** structure.

Figure R7. Free energy profiles for the conversion of $\text{O}_2 \rightarrow \text{O}_2^*$ and $\text{CH}_3\text{OH} + \text{O}_2^* \rightarrow \text{OOH}^* + \text{CH}_3\text{OH}^*$ on bulk ZnO and V_o -rich Fe/ZnO NSs.

With the reviewer's comments, we decided to add the above-revised Lines 222-224 to the manuscript for clarification. The revised text now reads:

Lines 256-263: *“Such transformation indicated the formation of alkoxy bonds between hydroxyl groups and Fe-Zn, thus providing evidence of the activation of hydroxyl groups in CH_3OH . Such results indicate that the β interaction (1056 cm^{-1}) and γ interaction (1004 cm^{-1}), attributed to the alkoxy bond between primary hydroxyl and metal oxides, gradually increase^{44,45}. Compared with the N_2 atmosphere (Supplementary Figure 14), the intensity of the hydroxyl group (1030 cm^{-1}) adsorbed on the catalyst surface remained unchanged, and the characteristic peaks of the β interaction (1056 cm^{-1}) and γ interaction (1004 cm^{-1}) did not appear, indicating that methanol was not activated by catalyst under nitrogen atmosphere.”*

Comment 16: Lines 227-229. This statement implies that Fe^{3+} donates d electrons to the bridging oxygen. It should be rewritten to clarify that electron transfer is from oxygen to Fe. Figure 3d also implies with the arrow direction that there is d -electron donation from Fe to O while the second statement in this sentence implies that there is electron donation from O to Fe.

Our Response: We sincerely thank the reviewer for the insightful comment. We redraw a new schematic (Supplementary Fig. 15) to clarify the electron transfer from oxygen to Fe in Figure 3d.

We have also revised Lines 227-229 of the manuscript for clarification. The revised text now reads:

Lines 265-269: *“The fully occupied π symmetry (t_{2g}) d -orbitals of Zn^{2+} interplay with the bridging O^{2-} by electron repulsion and half-occupied t_{2g} d -orbitals of Fe^{3+} interplay with the bridging O^{2-} by π donation, leading to partial electron transfer from Zn^{2+} to Fe^{3+} . It leads to better delocalization of the π symmetry electrons among Zn, and Fe in the host matrix (Figure*

~~3d). In $V_o\text{-Zn}^{2+}\text{-O-Fe}^{3+}$ sites, the three unpaired electrons in the π -symmetry (t_{2g}) d -orbitals of Fe^{3+} interact with the bridging O^{2-} via π -donation. In contrast, the dominant interaction between the fully occupied π -symmetry (t_{2g}) d -orbitals of Zn^{2+} and the bridging O^{2-} is electron-electron repulsion and half-occupied t_{2g} d -orbitals of Fe^{3+} interplay with the bridging O^{2-} by π -donation, leading to partial electron transfer from Zn^{2+} to Fe^{3+} (Supplementary Figure 15)^{46, 47}.~~

Revision Figure 3d to Figure S15 to the Supporting Information for clarification.

Supplementary Fig. 15. Schematic representations of the electronic coupling among bulk ZnO, and V_o -rich Fe/ZnO NSs.

Comment 17: Figure 3d shows four electrons in a single Zn d -orbital which, of course, cannot be the case. It is necessary to clarify what is meant by an oxygen vacancy. If it is meant that the $\text{Zn-}V_o\text{-Fe}$ site of interest is actually a Zn-O-Fe sites adjacent to the oxygen vacancy, as is suggested in Figure 3d, this must be made extremely clear to avoid the common confusion surrounding oxygen vacancies.

Our Response: We sincerely thank the reviewer for the insightful comment. Oxygen vacancies refer to defects left by oxygen atoms escaping from the crystal lattice in metal oxides or other oxygen-containing compounds. To address the confusion and the misrepresentation of the $\text{Zn-}V_o\text{-Fe}$ site in the manuscript, we have carefully reviewed and corrected the depiction of $V_o\text{-Zn}^{2+}\text{-O-Fe}^{3+}$ (Figure R7).

With the reviewer's comments, we have revised Figure 3d to ensure clarity and accuracy.

Figure R7. Schematic representations of the electronic coupling among bulk ZnO, and V_o -rich Fe/ZnO NSs.

Revision text in the manuscript:

Line 264: “We propose a $V_o\text{-Zn}^{2+}\text{-O-Fe}^{3+}$ site to analyze its electronic interplay of Zn, Fe ions, and oxygen vacancy. In $V_o\text{-Zn}^{2+}\text{-O-Fe}^{3+}$ sites, the three unpaired electrons in the π -symmetry

(t_{2g}) d -orbitals of Fe^{3+} interact with the bridging O^{2-} via π -donation. In contrast, the dominant interaction between the fully occupied π -symmetry (t_{2g}) d -orbitals of Zn^{2+} and the bridging O^{2-} is electron-electron repulsion and half-occupied t_{2g} d -orbitals of Fe^{3+} interplay with the bridging O^{2-} by π -donation, leading to partial electron transfer from Zn^{2+} to Fe^{3+} (Supplementary Figure 15)^{46, 47}.

Comment 18: Lines 230-231. It is not made clear how the claim that the proposed site favored the adsorption of the nucleophilic group is supported. Please state how this is known.

Our Response: We sincerely thank the reviewer for the insightful comment. We agree with the reviewer that “It is not made clear how the claim that the proposed site favored the adsorption of the nucleophilic group is supported.” The point of view of this sentence is correct, but the location of its appearance is confusing. We will explain this point of view in detail as follows:

We utilized the *in situ* attenuated total reflectance (ATR) Fourier-transform infrared spectra (FTIR) technique to study the changes in the functional groups of methanol at different reaction temperatures on the catalyst surface under an air atmosphere. Our analysis, illustrated in Figure R4, reveals the increase in the β interaction (1056 cm^{-1}) and γ interaction (1010 cm^{-1}), which are attributable to the formation of alkoxy bonds between the primary hydroxyl groups and metal oxides^{7,8} (*Nat. Commun.* 14, 4509 (2023); *Nat. Commun.* 13, 5467 (2022)). However, a decrease in $V_{as}(\text{OH})$ at $3600\text{--}3080\text{ cm}^{-1}$ indicates the activation of primary hydroxyl groups in methanol. This result suggests that methanol adsorbed on the surface of our V_o -rich-Fe/ZnO catalyst forms a specific structure (Structure 1, shown in Figure R5).

Figure R4. *In situ* attenuated total reflectance (ATR) infrared spectra investigated the changes in methanol functional groups adsorbed on the V_o -rich Fe/ZnO NSs surface under air atmosphere.

Figure R5. Schematic diagram of structure **1**.

The change of methanol activation energy barrier adsorbed on the V_o -rich Fe/ZnO NSs surface was further analyzed through theoretical calculation, as depicted in Figure R7. In V_o -Zn²⁺-O-Fe³⁺, the localized oxygen vacancy structure (denoted as **R**), serves as an anchoring site for an oxygen molecule, promoting the formation of the **M**₁ species. This interaction thus activates the hydroxyl hydrogen of the methanol (CH₃OH) molecule adsorbed on **M**₁, facilitated by the oxygen molecule, resulting in the formation of **M**₂ species. Then, the CH₃OH of **M**₂ interacts with the V_o -Zn²⁺-O-Fe³⁺ to form a transition state (denoted as **TS**) characterized by a metal alkoxy bond, which eventually leads to the formation of nucleophilic species in the **M**₃ structure.

Figure R7. Free energy profiles for the conversion of $O_2 \rightarrow O_2^*$ and $CH_3OH + O_2^* \rightarrow OOH^* + CH_3OH^*$ on bulk ZnO and V_o -rich Fe/ZnO NSs.

Comment 19: Lines 232-233. The authors claim that O_2 activation is an important reaction for methoxy formation. However, the reactions are run under a N_2 atmosphere, so it is confusing that O_2 is invoked. An alternative route should be explored that does not rely on O_2 .

Our Response: We sincerely thank the reviewer for the insightful comment. We have corrected the confusion statement regarding the evaluation of the catalytic performance of Lines XX-XX in the manuscript. The polyester depolymerization reaction in the reactor is carried out under the air or N₂.

Typically, alcoholysis reactions are carried out under nitrogen protection. However, our research demonstrates that V_o-Fe/ZnO NSs can completely convert PET under an oxygen atmosphere (through purging air into the reactor), leading to a high yield of monomer DMT (> 99%). In comparison under nitrogen protection, the PET conversion rate is less than 10%, with the products being mainly oligomers. These supplementary experimental results are provided in Figure R10.

Figure R10. The polyester depolymerization reaction is carried out under the N₂.

To address the reviewer's comments, we have revised the text in Lines 232-233 for clarification. The revised text now reads:

Lines 515-519: ~~“The autoclave was purged with N₂ to eliminate residual air at ambient temperature. The polyester depolymerization reaction in the reactor is carried out under the air. To verify the effect of reaction atmosphere on depolymerization performance, the polyester depolymerization reaction was purged with N₂ to eliminate residual air at ambient temperature”~~

Added a new Figure S5 to the Supporting Information:

Supplementary Fig. 5. The polyester depolymerization reaction is carried out under the N₂.

Comment 20: Line 243. The authors state that methanol is dehydrogenated when it is only deprotonated. Methanol dehydrogenation would lead to formaldehyde, not methoxy species.

Our Response: Thank you for your constructive suggestions. We appreciate the opportunity to clarify this point. We utilized the *in situ* attenuated total reflectance (ATR) Fourier-transform infrared spectra (FTIR) technique to study the changes in the functional groups of methanol at different reaction temperatures on the catalyst surface under an air atmosphere. Our analysis, illustrated in Figure R4, reveals the increase in the β interaction (1056 cm^{-1}) and γ interaction (1010 cm^{-1}), which are attributable to the formation of alkoxy bonds between the primary hydroxyl groups and metal oxides^{7,8}(*Nat. Commun.* 14, 4509 (2023); *Nat. Commun.* 13, 5467 (2022)). However, a decrease in $V_{as}(\text{OH})$ at $3600\text{--}3080\text{ cm}^{-1}$ indicates the activation of primary hydroxyl groups in methanol. This result suggests that methanol adsorbed on the surface of our V_o -rich-Fe/ZnO catalyst forms a specific structure (Structure **1**, shown in Figure R5).

Figure R4. *In situ* attenuated total reflectance (ATR) infrared spectra investigated the changes in methanol functional groups adsorbed on the V_o -rich Fe/ZnO NSs surface under air atmosphere.

Figure R5. Schematic diagram of structure **1**.

The change of methanol activation energy barrier adsorbed on the V_o -rich Fe/ZnO NSs surface was further analyzed through theoretical calculation, as depicted in Figure R7. In $V_o\text{-Zn}^{2+}\text{-O-Fe}^{3+}$, the localized oxygen vacancy structure (denoted as **R**), serves as an anchoring site for an oxygen molecule, promoting the formation of the **M₁** species. This interaction thus activates the hydroxyl hydrogen of the methanol (CH_3OH) molecule adsorbed on **M₁**, facilitated by the oxygen molecule, resulting in the formation of **M₂**

species. Then, the CH_3OH of \mathbf{M}_2 interacts with the $\text{V}_o\text{-Zn}^{2+}\text{-O-Fe}^{3+}$ to form a transition state (denoted as \mathbf{TS}) characterized by a metal alkoxy bond, which eventually leads to the formation of nucleophilic species in the \mathbf{M}_3 structure.

Figure R7. Free energy profiles for the conversion of $\text{O}_2 \rightarrow \text{O}_2^*$ and $\text{CH}_3\text{OH} + \text{O}_2^* \rightarrow \text{OOH}^* + \text{CH}_3\text{OH}^*$ on bulk ZnO and V_o -rich Fe/ZnO NSs.

Comment 21: Lines 260 and 270. Colored shaded boxes are referred to in Figure 4, but these are not present. Figure 4c. This figure shows a branched polyethylene polymer it appears. It should reflect the structure of PET better by being linear.

Our Response: We acknowledge that the shaded portions of Figure 4 were obscured by the gray background. In response, we have redrawn Figure 4 to include visible shaded boxes, as shown in Figure R11. Additionally, we agree that the polymer structure depicted in the figure was not accurate. To address this issue, we have also redrawn the polymer structure to reflect the linear structure of PET, as shown in Figure R12.

Figure R11. GPC profiles of different reaction times over PET depolymerization.

Figure R12. Different types of bond scission occur in PET depolymerization.

Following the reviewer's comments, we have revised Figure 4 in the manuscript for clarification.

Figure 4. a, b DSC and GPC profiles of different reaction times over PET depolymerization. c ¹H NMR spectrum of Modes 1 and 2 in DMSO-d₆. d Mass spectrogram of Modes 1 and 2 depolymerization.

Added a new figure (Figure S18) to the Supporting Information:

Supplementary Fig. 18. Different types of bond scission occurring in PET depolymerization.

Comment 22: *The entirety of the isotopic labeling section is lacking in explanation and context. As it is currently, it is difficult to gain any insights from the experiments.*

Our Response: In response to the reviewer's comment, we would like to provide a more detailed explanation of the background and purpose of the isotope experiments.

Isotope experiments were conducted to investigate the role of methanol in the conversion of dimers into monomers. We synthesized two model compounds to replace the dimer and studied the bond-breaking positions when attacked by methanol, as illustrated in Figure R13. When methanol is used as the solvent, **Mode 1** depolymerization occurs to yield DMT (Figure R13a-b), corresponding to the signal at $m/z=194$ (Figure R13e). Using deuterated methanol as the solvent, the depolymerization product is D_3 -DMT, corresponding to the signal at $m/z=197$ (Figure R13e). These results indicate that the bond-breaking reaction occurred at the $O=C-O-CH_2$ position of the dimer, rather than the $O=C-O-CH_3$ position. Similarly, **Mode 2** depolymerization is generated when methanol is the solvent, showing the signal at $m/z=194$ (Figure R13e). When deuterated methanol is used, the depolymerization product is D_3 -DMT (Figure R13c-d), corresponding to the signal at $m/z=197$ (Figure R13e). Such a result further verified that the bond breaking position of the dimer is $O=C-O-CH_2$.

Taken together, we believe these results from the isotopic labeling experiment provide valuable insights into the specific bond breaking mechanisms during the methanolysis of PET, contributing to a deeper understanding of the depolymerization process.

Figure R13. a-d Methanolysis of Modes 1 and 2 under methanol and methanol-d₄. e Mass spectrogram of Modes 1 and 2 depolymerization.

With the reviewer's comments, we have the relevant section in the manuscript including this detailed explanation for better clarity and context. The revised text now reads:

Lines 327–328: “We further performed isotope-labeling experiments ~~to conclusively determine the source of the produced DMT.~~ we further conducted to investigate the action of methanol in the conversion of dimer into a monomer reaction.”

Supplementary Fig. 21. a-d Methanolysis of Modes 1 and 2 under methanol and methanol-d₄.

Comment 23: *Figure 4d. The structures in the middle panel have an extra carbon in the esters.*

Our Response: We deeply appreciate the reviewer's effort in pointing out this error in the figure. We have corrected the error in Figure 4d of the manuscript, where the structures in the middle panel had extra carbon in the esters. The corrected figure is shown in Figure R14. The relevant content has also been moved to the supplementary information. Additionally, we have reviewed the corresponding **Mode** structure in the supplementary information to ensure its accuracy and completeness.

Figure R14. Synthetic route of **Modes 1** and **2**.

In revision, we have decided to add new figure (Figure S19) to the Supporting Information:

Supplementary Fig. 19. Synthetic route of **Modes 1** and **2**.

Revised Figure S20 to the Supporting Information:

Supplementary Fig. 20. ^1H NMR spectra of **Modes 1** and **2** recorded in CD_3Cl .

Comment 24: Lines 299–300. The authors should be careful in their use of the terms radicals and deprotonation. Deprotonation is a heterolytic process whereas radical generation would be homolytic in this case. This terminology should be updated to reflect the chemistry more accurately.

Our Response: We agree with the reviewer that the statement of CH_3O^* and H^* radicals in the original article is inaccurate. We have now revised the manuscript to accurately reflect the chemistry. To revised text now reads:

Lines 339–342: “This finding verifies that ~~CH_3O^* and H^* radicals generated from the deprotonation of CH_3OH facilitates triggering the cleavage of $\text{C}-\text{O}$ bond in **Modes 1** and **2** bond breaking position is at the $\text{O}=\text{C}-\text{O}-\text{CH}_2$ position of the dimer, not the $\text{O}=\text{C}-\text{O}-\text{CH}_3$ position, thus leading to the formation of $\text{H}_3\text{CO}(\text{O})\text{C}-\text{C}_6\text{H}_4-\text{C}(\text{O})\text{OCD}_3$ and $\text{H}_3\text{CO}(\text{O})\text{C}-\text{C}_6\text{H}_4-\text{C}(\text{O})\text{OCH}_3$.”~~

Comment 25: Lines 453–470. The source and (particle) size of PET used for optimization reactions need to be specified. Additionally, the HPLC (for glycolysis reactions) and GC (for methanolysis reactions) methods should be added to the SI with an example chromatograph.

Our Response: Thank the reviewer for the valuable suggestions. Based on the reviewers' comments, we have added the details of the analysis method for depolymerization products to the supplementary material method.

Supporting Information, Pages S2–S3, Lines 60–68: “**High-performance liquid chromatography (HPLC) analysis:** The methanolysis products were analyzed using an Agilent 8860 HPLC system equipped with a C18 column and an ultraviolet (UV) detector set at 254 nm. A 50:50 (v/v) methanol/ H_2O mixed solution was used as the mobile phase at a flow rate of 1.0 mL/min.

Gas chromatography (GC) analysis: The methanolysis products were analyzed by an Agilent 8860 GC system equipment with an HP-5 column (30 m×0.25 mm). The column temperature was initially set at 50 °C (held for 5 minutes) and then increases to 250 °C (held for 5 minutes) at 10 °C·min⁻¹. An internal standard, n-heptane, was added to the solution and homogeneously mixed after the reaction.”

Comment 26: *Depolymerization reactions were carried out after purging of air with nitrogen, but in situ IR experiments were carried out under air. Since O₂ is thought to be an active species, this is an important difference and in situ experiments should be run under nitrogen.*

Our Response: Thank you for your valuable suggestions. Based on the reviewers' comments, we have added *in situ* IR experiments that were carried out under nitrogen (Figure 15).

Then we synthesized **Modes 1 and 2** to simulate dimers and used *in situ* FTIR to study the changes in functional groups during the depolymerization of these **Modes** into monomers within the **M₃** structure. The specific analysis utilized *in situ* high-temperature-pressure infrared spectrometric (*in situ* HTP-IR) techniques were carried out at a synchrotron source, as detailed in Figure R8. This approach enabled us to delineate the depolymerization pathways by detecting intermediates and assessing isotope effects during the reactions.

As shown in Figure 5b, depolymerization in **Mode 1** with CH₃OH efficiently produces various intermediate species, including:

- 1) $\nu(\text{C}=\text{O})$ at 1730 cm⁻¹;
- 2) $\nu(\text{C}-\text{O})$ at 1440 and 1259 cm⁻¹;
- 3) $\nu(\text{O}-\text{C}-\text{H})$ at 1338 cm⁻¹;
- 4) $\nu(\text{C}-\text{OH})$ at 1130 cm⁻¹;
- 5) $\nu(\text{C}-\text{H})$ at 725 cm⁻¹.

The conversion of a hydroxyl group (1083 cm⁻¹) into triply bridged hydroxyl groups (1022, 1080, and 1118 cm⁻¹) indicates the formation of alkoxy bonds between hydroxyl groups and the catalyst (Figure 5b). The increasing intensity of $\nu(\text{C}-\text{O})$ at 1440 cm⁻¹ and $\nu(\text{O}-\text{C}-\text{H})$ at 1338 cm⁻¹ suggests the activation of the hydroxyl group in CH₃OH. Moreover, the peak intensities of these intermediates (C=O, C-O, and C-H) gradually rise, implying the transformation of **Mode 1** into DMT during the methanolysis process.

In comparison, *in situ* FTIR spectra showed that the OD absorbance band at 2500 cm⁻¹ progressively increased and eventually stabilized upon adding CD₃OD (Figure 5c). As the reaction progresses, the OH bonds (3400–3700 cm⁻¹) on the surface are progressively increased, leading to the generation of ethylene glycol (EG) *via* **Mode 1** depolymerization. Taken together, these findings underscore the role of vacancies in promoting oxygen dissociation and transfer, which are critical processes in the formation of the **M₃** structure

formation. Consequently, the interaction of Zn and CH_3OH^* within the M_3 structure activates the carbonyl groups ($\text{O}=\text{C}-\text{O}$) of **Mode 1** and promotes $\text{C}-\text{O}$ bond cleavage, resulting in the formation of DMT and ethylene glycol.

Figure R8. *In situ* high-temperature-pressure infrared (*in situ* HTP-IR) spectra were investigated by **Mode 1** depolymerization in the V_o -rich Fe/ZnO NSs surface under air and N_2 atmosphere.

With the reviewer's comments, we decided to add the above-revised Lines 360–368 to the manuscript for clarification.

Lines 360-368: “...As the reaction progresses, the OH bonds ($3400\text{--}3700\text{ cm}^{-1}$) on the surface are progressively increased to the generation of EG by **Mode 1** depolymerization (Supplementary Figure 22a, b). ~~These results suggest that the V_o -rich Fe/ZnO NSs efficiently convert CH_3OH into the CH_3O^* intermediate via dehydrogenation, thus accelerating the depolymerization of PET. It is worth noting that vacancies on catalysts could facilitate hydrogen dissociation and transfer, as demonstrated in the literature.~~ In contrast, under the nitrogen atmosphere, the *in situ* IR spectrum of the depolymerization reaction does not exhibit the characteristic functional group changes mentioned above (Supplementary Figure 23). Taken together, these vacancies can promote oxygen dissociation and transfer, which are critical for the formation of the M_3 structure. Thus, Zn and CH_3OH^* on the M_3 structure activate the carbon groups ($\text{O}=\text{C}-\text{O}$) of **Mode 1** and promote $\text{C}-\text{O}$ bond breaking to form DMT and ethylene glycol.”

Added a new figure (Figure S23) to the Supporting Information:

Supplementary Fig. 23. In situ high-temperature-pressure infrared spectrometry was investigated by Mode 1 depolymerization under nitrogen.

Comment 27: *There should be ICP reported for the catalyst synthesis. The provided TEM image does not clearly show vacancies, especially not 10% of the surface metal as would be suggested by the ratio of Zn to Fe in the synthesis without further characterization.*

Our Response: Following the reviewers' comments, we have added the characterization of the catalyst using EDS and ICP. The results show that the Fe content of the catalyst is 0.7% and 0.8%, respectively (Figure R16, Supplementary Figure 9). Therefore, the catalyst can be defined as V_o -rich-0.8%Fe/ZnO NSs.

Additionally, based on the reviewers' comments, we have observed the existence of vacancies on the surface of V_o -rich-0.8%Fe/ZnO NSs using aberration-corrected high-angle annular dark-field scanning TEM. As displayed in Figure R17, slight lattice disorders have been locally observed in the nanosheets, which are likely the result of the vacancies induced by the unsaturated coordination of metal atoms.

The oxygen vacancies were further resolved by electron spin resonance (ESR) spectroscopy (Figure R18), a tool for examining unpaired electrons in materials. The V_o -rich Fe/ZnO NSs exhibit a symmetrical ESR signal at $g = 2.002$, manifesting the electron trapping at oxygen vacancies. To further explore their defect structure, X-ray photoelectron spectroscopy (XPS) was employed to reveal the valence states of the samples. The O 2p peak located at 529.7 eV is attributed to the lattice oxygen, while the other one located at 531.4 eV corresponds to oxygen atoms in the vicinity of oxygen vacancies (Figure 19).

Figure 16. **a** EDS and ICP tables of V_o -rich Fe/ZnO NSs, **b** EDS image of V_o -rich Fe/ZnO NSs.

Figure R17. Aberration-corrected HAADF-STEM image of V_o -rich Fe/ZnO NSs.

Figure R18. ESR profiles of V_o -rich Fe/ZnO NSs.

Figure R19. O 2p XPS spectra of V_o -rich Fe/ZnO NSs.

Following the reviewer's suggestions, we have added the above results and analysis to the manuscript for clarification. The revised text now reads:

Lines 193-215: “*Aberration-corrected high-angle annular dark-field scanning transmission electron microscopy (HAADF-STEM) was employed to reveal the fine structures of V_o -rich Fe/ZnO nanosheets (NSs). As depicted in Figure 2d, slight lattice disorders have been locally observed in the nanosheets, likely stemming from vacancies induced by the unsaturated coordination of metal atoms. The V_o -rich Fe/ZnO NSs show interplanar spacings of 0.283 nm, corresponding to the distances of the (100) planes of ZnO (Supplementary Figure 9). Energy-dispersive X-ray spectroscopy (EDS) and inductively coupled plasma (ICP) analysis indicate that the iron doping content of the nanosheets is 0.7% and 0.8%, respectively (Supplementary*

Figure 10). This iron concentration is close to the theoretical value (~1%) determined from the weight ratio used during the catalyst synthesis. Unpaired electrons on these defects could be further assessed using electron spin resonance (ESR) (Figure 2e). Bulk ZnO without vacancies showed an intensity at a g value of 1.950, attributed to electron trapping at the lattice of Zn sites (V_{Zn})⁴². In comparison, V_{o} -rich Fe/ZnO NSs have abundant oxygen vacancies, as evidenced by their strong ESR intensity at a g value of 2.002. We used X-ray photoelectron spectroscopy (XPS) to determine the valence state of $V_{\text{o}}\text{-Zn}^{2+}\text{-O-Fe}^{\delta+}$. In the Zn 2p region (Supplementary Figure 11a), Zn 2p_{1/2} and Zn 2p_{3/2} peaks appear at 1044.7 and 1021.6 eV, respectively, indicative of the +2 oxidation state of Zn. In the Fe 2p region (Supplementary Figure 11b), Fe 2p_{1/2} and Fe 2p_{3/2} peaks appear at 724.9 and 711.3 eV, indicative of the +3 oxidation state of Fe. As such, $V_{\text{o}}\text{-Zn}^{2+}\text{-O-Fe}^{\delta+}$ was determined to be $V_{\text{o}}\text{-Zn}^{2+}\text{-O-Fe}^{3+}$. Additionally, in the O 1s region (Supplementary Figure 11c), the peaks at 531.4 eV and 529.7 eV correspond to the O atoms in the vicinity of oxygen vacancies and the lattice oxygen of Zn–O–Fe, respectively⁴².”

Revised Figure 2d to the manuscript:

Figure 2d. Aberration-corrected HAADF-STEM image of V_{o} -rich Fe/ZnO NSs.

Added a new figure (Figure S9–S11) to the Supporting Information:

Supplementary Fig. 9. HRTEM image of V_{o} -rich Fe/ZnO NSs.

Supplementary Fig. 10. a EDS and ICP tables of V_o -rich Fe/ZnO NSs, b EDS image of V_o -rich Fe/ZnO NSs.

Supplementary Fig. 11. High-resolution XPS spectra of the as-prepared V_o -rich Fe/ZnO NSs catalysts. XPS spectra of a Zn 2p, b Fe 2p, and c O 2p.

References

1. Arifuzzaman, M. et al. Selective deconstruction of mixed plastics by a tailored organocatalyst. *Mater Horiz* **10**, 3360-3368 (2023)
2. Zhang, Z. et al. Mixed Plastics Wastes Upcycling with High-Stability Single-Atom Ru Catalyst. *J. Am. Chem. Soc.* **145**, 22836-22844 (2023).
3. Clark, R. A. & Shaver, M. P. Depolymerization within a Circular Plastics System. *Chem Rev* **124**, 2617-2650 (2024).
4. Cao, J. et al. Mechanism of the Significant Acceleration of Polyethylene Terephthalate Glycolysis by Defective Ultrathin ZnO Nanosheets with Heteroatom Doping. *ACS Sustainable Chem. Eng.* **10**, 5476-5488, (2022).
5. Cao, J. et al. Molecular oxygen-assisted in defect-rich ZnO for catalytic depolymerization of polyethylene terephthalate. *iScience* **26**, 107492 (2023).
6. Yun, L.-X., Wu, H., Shen, Z.-G., Fu, J.-W. & Wang, J.-X. Ultrasmall CeO₂ Nanoparticles with Rich Oxygen Defects as Novel Catalysts for Efficient Glycolysis of Polyethylene Terephthalate. *ACS Sustainable Chem. Eng.* **10**, 5278-5287 (2022).
7. Yan, H. et al. Enhancing polyol/sugar cascade oxidation to formic acid with defect rich MnO₂ catalysts. *Nat. Commun.* **14**, 4509 (2023).
8. An, Z. et al. Pt₁ enhanced C-H activation synergistic with Pt_n catalysis for glycerol cascade oxidation to glyceric acid. *Nat. Commun.* **13**, 5467 (2022).
9. McLaren, A., Valdes-Solis, T., Li, G. & Tsang, S. C. Shape and Size Effects of ZnO Nanocrystals on Photocatalytic Activity. *J. Am. Chem. Soc.* **131**, 12540-12541 (2009).
10. Peng, Y.-K. et al. Trimethylphosphine-Assisted Surface Fingerprinting of Metal Oxide Nanoparticle by ³¹P Solid-State NMR: A Zinc Oxide Case Study. *J. Am. Chem. Soc.* **138**, 2225-2234 (2016).

REVIEWERS' COMMENTS

Reviewer #1 (Remarks to the Author):

I was supportive of publication with some minor additions - which the authors have undertaken and well addressed. I must say the level of the response to the reviewers is incredibly high and the authors should be commended on the thorough job that they have done. My opinion is that the manuscript is now acceptable in the current form.

Reviewer #2 (Remarks to the Author):

The authors have addressed all the comments and suggestions raised by reviewers in the first round of analysis. The current version of the document is free from any weaknesses of expression, conceptual defects, or dichotomies that could discredit this work. It demonstrates originality and effectively contributes to the area of research in which it is developed. Therefore, I recommend publishing this manuscript in Nature Communications. Congratulations to the authors on their meticulous work.

Reviewer #3 (Remarks to the Author):

The authors did respond to all of my comments and seem to have improved the manuscript a lot.